# A fungal substrate mimicking molecule suppresses plant immunity via an inter-kingdom conserved motif

Johana C. Misas Villamil[1], André N. Mueller[2], Fatih Demir [3], Ute Meyer[1], Bilal Ökmen[1], Jan Schulze Hüynck[1], Marlen Breuer[2], Helen Dauben[1], Joe Win[4], Pitter F. Huesgen [3,5] & Gunther Doehlemann [1]

*Ustilago maydis* is a biotrophic fungus causing corn smut disease in maize. The secreted effector protein Pit2 is an inhibitor of papain-like cysteine proteases (PLCPs) essential for virulence. Pit2 inhibitory function relies on a conserved 14 amino acids motif (PID14). Here we show that synthetic PID14 peptides act more efficiently as PLCP inhibitors than the full-length Pit2 effector. Mass spectrometry shows processing of Pit2 by maize PLCPs, which releases an inhibitory core motif from the PID14 sequence. Mutational analysis demonstrates that two conserved residues are essential for Pit2 function. We propose that the Pit2 effector functions as a substrate mimicking molecule: Pit2 is a suitable substrate for apoplastic PLCPs and its processing releases the embedded inhibitor peptide, which in turn blocks PLCPs to modulate host immunity. Remarkably, the PID14 core motif is present in several plant associated fungi and bacteria, indicating the existence of a conserved microbial inhibitor of proteases (cMIP).

---

[1] Botanical Institute and Cluster of Excellence on Plant Sciences, University of Cologne, Cologne D-50674, Germany. [2] Max Planck Institute for Terrestrial Microbiology, Marburg D-35043, Germany. [3] Central Institute for Engineering, Electronics and Analytics, ZEA-3, Forschungszentrum Jülich, Jülich D-52425, Germany. [4] The Sainsbury Laboratory, Norwich Research Park, Norwich NR4 7UH, UK. [5] Medical Faculty and University Hospital Cologne, University of Cologne, Cologne D-50931, Germany. These authors contributed equally: Johana Misas Villamil, André N. Mueller. Correspondence and requests for materials should be addressed to G.D. (email: g.doehlemann@uni-koeln.de)

Food security and global food supply has become one of the major challenges to address this century as the world population increases. Plant diseases caused by microbial pathogens are responsible for ca. 16% crop-losses worldwide, being fungal pathogens accountable for the most devastating diseases[1,2]. To improve crop resistance it is necessary to understand how pathogens colonize their host and the role of pathogen effectors at the functional level. Only few eukaryotic plant pathogen model systems can be used to elucidate the function of fungal effectors. One of those examples is the biotrophic fungus *Ustilago maydis* responsible for corn smut disease in maize. To manipulate its host *U. maydis* secretes a set of effectors into the extracellular interface aiming to downregulate immune responses thus achieving successful colonization[3]. Effectors can be translocated into the plant cell where they can modulate host cell biology. Cmu1, for example, is a cytosolic localized chorismate mutase effector which functions by manipulating host signaling cascades and has been postulated to decrease salicylic acid levels in the cell, priming plant cells for an upcoming infection[4,5]. Another translocated effector manipulating signaling cascades is Tin2, which has been postulated to decrease the biosynthesis of lignin as a strategy to channel metabolites into the anthocyanin pathway reducing their availability for other defense pathways[6]. See1 is an organ specific effector translocated into the plant cell nucleus and cytoplasm where it reprograms the cell cycle progression of host cells contributing to tumor formation[7,8]. Effectors can have a function in the apoplast that is believed to dismantle plant defense, such as Pep1 that inhibits host peroxidases avoiding the accumulation of reactive oxygen species[9]. Recently, the repetitive effector protein Rsp3 has been shown to bind the *U. maydis* cell wall working as a shield of fungal hyphae, thus blocking the antifungal activity of mannose-binding maize proteins[10]. The Pit2 effector inhibits apoplastic cysteine proteases, which are key regulators of plant defense directly linked to salicylic acid signaling in maize[11,12].

Papain-like cysteine proteases (PLCPs) are hubs in plant immunity[13]. PLCPs are cysteine proteases belonging to the C1A family of clan CA represented by papain as the family type peptidase that have the conserved catalytic triad Cys, His, Asn[14]. During plant immune signaling PLCPs might have different functions[13]. For example, PLCPs may release pathogen/microbe-associated molecular patterns (PAMPs/MAMPs) or damage-associated molecular patterns (DAMPs), which are recognized by plant receptors activating immune responses. In maize, the peptide Zip1 has recently been found being induced after SA treatment and released from a propeptide precursor requiring PLCP activity[15]. Zip1 triggers SA signaling in maize leaves leading to an activation of PLCPs, thus establishing a feedback loop in order to promote induction of immune responses[15,16]. Some PLCPs can act as coreceptors and decoys conferring resistance against different pathogens such as the tomato Rcr3 required for the function of the receptor-like protein Cf-2. The complex Rcr3-Cf2 enhances resistance against the fungal pathogen *Cladosporium fulvum* and the nematode *Globodera rostochiensis*, which secrete the Rcr3 inhibitors Avr2 and Gr-VAP1, respectively[17,18]. Interestingly, PLCPs are mostly posttranscriptionally regulated and the plant cell controls their activity with the production of endogenous inhibitors such as cystatins, serpins or kunitz[19–21]. This posttranslational regulation might be used by different pathogens since PLCPs are common targets of pathogen effectors. Unrelated plant pathogens such as bacteria, fungi, oomycetes, nematodes and viruses produce inhibitors of PLCPs in order to interfere with their activity and subcellular localization[13].

Pit2 has been described as a core effector in smut fungi[22] and is conserved in the four smut genomes of *Ustilago hordei*, *Ustilago maydis*, *Sporisorium reilianum* and *Melanopsichium*

*pennsylvanicum*[23]. Pit2 inhibitory function of PLCPs has been only described in *U. maydis*[12] and as many other pathogen PLCP inhibitors its structure is still unknown. Besides, Pit2 does not have recognized domains or sequence similarities with known enzyme inhibitors.

Here we elucidate the mechanism of action of the essential virulence effector Pit2. We found that orthologs of Pit2 from related smut pathogens cannot complement tumor formation in maize. We show that the function of Pit2 might be host specific, although the PLCP inhibitory PID14 motif is conserved in unrelated fungi and bacteria. Based on our findings, we propose that *U. maydis* Pit2 acts as a substrate mimicking molecule. Molecular mimicries are molecules produced by pathogens, which resemble host factors such as enzyme substrates to suppress host immune responses, facilitate infection and maintain the biotrophic interaction[24]. Pit2 is recognized as a substrate of host PLCPs. Cleavage by the proteases releases the inhibitory portion, which results in inactivation of the PLCPs and consequently the suppression of plant immunity. Both features, substrate recognition by PLCPs and subsequently PLCP inhibition, are required for full Pit2 virulence function in maize.

## Results

**Pit2 orthologs cannot complement tumor formation in maize.** Although the *pit2* gene was found to be present in the four smuts, *U. maydis*, *U. hordei*, *S. reilianum* and *M. pennsylvanicum*, previous analyses have shown high sequence divergence for this gene[23] (Fig. 1a). To address the question if Pit2 orthologs have the same function, a complementation experiment using the *U. maydis pit2* deletion strain SG200_Δpit2 was performed. The SG200_Δpit2 mutant strain was genetically complemented with *srpit2*, *mppit2* and *uhpit2* and disease symptoms in maize leaves caused by the resulting strains were rated at 12 days-post-infection (dpi). While *srpit2* and *mppit2* resulted in a partial complementation of virulence, *uhpit2* could not complement tumor formation phenotype (Fig. 1b, c). The latter finding was surprising, as the UhPit2 protein contains the conserved protease inhibitory domain (PID14), which previously was shown to be required for Pit2 function[12]. To analyze if the PID14 motif of both *U. maydis* and *U. hordei* have a similar function during infection, *U. maydis* strains expressing secreted versions of either UhPID14 or UmPID14 were generated and used for maize infections. Production and secretion of Pit2 variants and PID14 peptides in the recombinant *U. maydis* strains was confirmed by confocal microscopy (Supplementary Fig. 1). In accordance with previous reports[11,12], the genetic complementation of strain ΔPit2 with wild-type UmPit2 shows full restoration of virulence when compared to the *U. maydis* progenitor strain SG200 strain (Fig. 2a). Secretion of only UmPID14 in the ΔPit2 mutant background results in partially restored virulence, demonstrating a virulence function of the UmPID14 peptide (Fig. 2a, b). In contrast, UhPID14 did not rescue the virulence defect in the ΔPit2_UhPID14 mutant (Fig. 2a, b). To confirm the virulence function of UmPID14, Pit2 chimeras of both UhPID14 and UmPID14 were generated (Fig. 2c). Disease rating experiments with the generated effector chimeras showed that UmPit2-UhPID14 cannot rescue tumor formation on the SG200_ΔPit2 strain, while UhPit2-UmPID14 partially restored virulence (Fig. 2d). These experiments demonstrate that UmPID14 itself is a virulence factor essential for Pit2 function in maize and suggest a different function for Pit2 in both *U. maydis* and *U. hordei* smuts.

To elucidate the lack of function found for UhPit2, we compared UhPit2 and UmPit2 inhibition of apoplastic cysteine proteases. An in vitro concentration range using heterologous

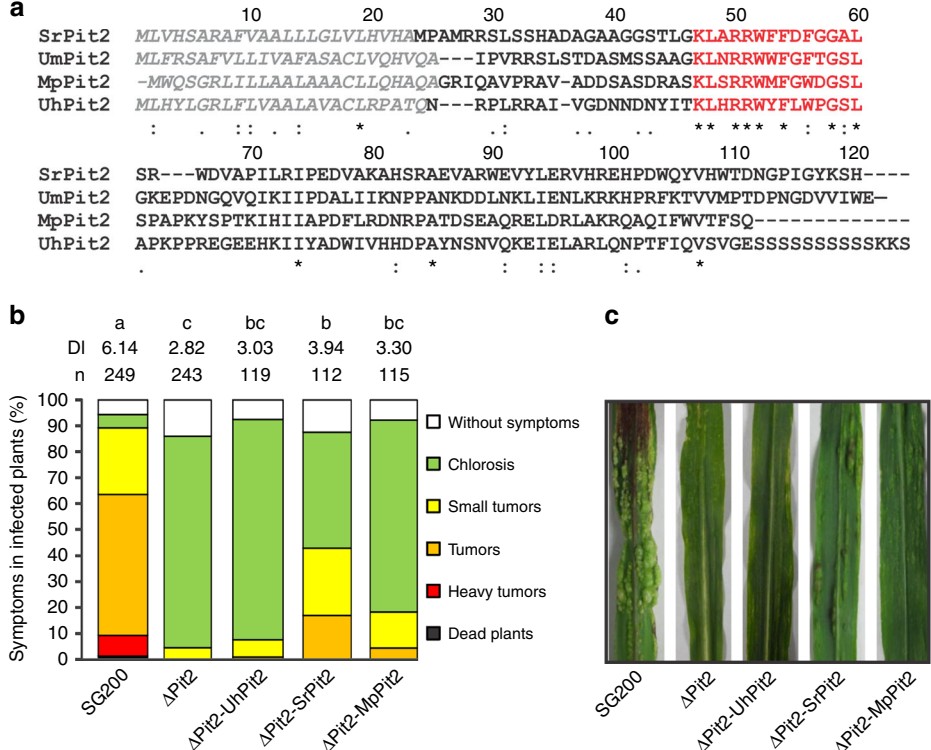

**Fig. 1** Pit2 homologs cannot complement tumor formation phenotype. **a** Sequence alignment of Pit2 orthologs from *Sporisorium reilianum* (Sr), *Ustilago maydis* (Um), *Melanopsichium pennsylvanicum* (Mp) and *Ustilago hordei* (Uh). Signal peptide (gray) and PID14 conserved motif (red) are depicted. Amino acids are labeled depending on their conservation: identical amino acids (asterisk), similar hydrophobicity (colon) or different hydrophobicity (dot). **b** Disease rating of maize seedlings 12 days post infection (dpi) with *U. maydis* SG200 solopathogenic strain, SG200 mutant strain lacking Pit2 (ΔPit2) and complemented ΔPit2 strain with *U. hordei* Pit2 (ΔPit2-UhPit2), *S. reilianun* Pit2 (ΔPit2-SrPit2) and *M. pennsylvanicum* Pit2 (ΔPit2-MpPit2). DI: disease index calculated as a mean of at least three biological replicates. Letters above indicate significant differences between samples (α = 0.05, Tukey test). **c** Disease symptoms of maize seedlings at 12 dpi infected with strains used in (**b**)

expressed Pit2 proteins and maize apoplastic fluids containing immune-activated PLCPs was performed. Inhibitory profiles show that in comparison to UmPit2, UhPit2 is a less efficient inhibitor of maize PLCPs (Fig. 3a). This is also reflected by PLCP activity in maize leaves 3 days post infection (dp1). The activity of PLCPs was significantly induced in leaves infected by the SG200_ΔPit2 mutant when compared to samples infected by the SG200, or the ΔPit2-UmPit2-complemented strain (Fig. 3b). In contrast, the ΔPit2-UhPit2-complemented strain shows an intermediate phenotype, which is in line with the lower affinity of UhPit2 to maize PLCPs compared to UmPit2 (Fig. 3b). To determine if UhPID14 inhibits maize PLCPs, synthetic peptides were tested against PLCPs from maize apoplastic fluids. This experiment showed that UhPID14 can inhibit maize PLCPs, but with a significantly lower efficiency when compared to UmPID14 (Fig. 3c). UhPID14 inhibitory profile of heterologous expressed CP1 and CP2 maize proteins shows that both proteins can be inhibited at higher concentrations but with a low efficiency (Supplementary Fig. 2). These results confirm the poor efficiency of UhPit2 as an inhibitor of maize PLCPs in vitro and during maize infection.

**UmPit2 is a substrate of maize PLCPs**. Despite the relatively low affinity of UhPit2 to maize PLCPs, its deficiency in complementing *U. maydis* tumor formation could be caused by an eventual instability of the heterologous effector in the maize apoplast. To address this issue, both recombinant UmPit2 and

UhPit2 proteins were incubated over time with maize apoplastic fluid containing active PLCPs. Surprisingly, after 15 min incubation UmPit2 was largely processed, whereas UhPit2 appeared to be stable (Fig. 4a, b). Incubation of UhPit2 and UmPit2 in the control samples without apoplastic fluid did not show protein degradation (Fig. 4a, b). This experiment suggests that UmPit2 is processed in the apoplast, whereas UhPit2 remains stable. To determine if processing of UmPit2 in the apoplast is due to protease activity, four inhibitors targeting specific group of proteases, which display the major activities found in the plant apoplast, were tested: E-64 to inhibit PLCPs, EDTA to inhibit metalloprotease activity, 3,4-dichloroisocoumarin (DCI) to inhibit serine protease activity and pepstatin A to suppress the activity of aspartic proteases. Control samples of UmPit2 incubated without apoplastic fluid showed no degradation of UmPit2 during the time-course of the experiment (Fig. 5a). Processing of UmPit2 was observed gradually in apoplastic fluid preincubated with EDTA, DCI and pepstatin-A, but not in apoplastic fluid pre-incubated with E-64 (Fig. 5b), indicating that E-64 stabilizes UmPit2 and thereby suggesting that PLCPs are responsible for processing of UmPit2. A coincubation of apoplastic fluid with a mixture of inhibitors containing EDTA, DCI and pepstatin-A did not prevent processing of UmPit2 similar to the control without inhibitors, whereas addition of E-64 to the mixture resulted in stabilization of UmPit2 (Fig. 5c). These experiments led us to conclude that UmPit2 acts as a substrate of maize PLCPs. An enrichment of immune-active PLCPs was performed to confirm PLCPs as the actual proteases cleaving UmPit2 in the apoplast.

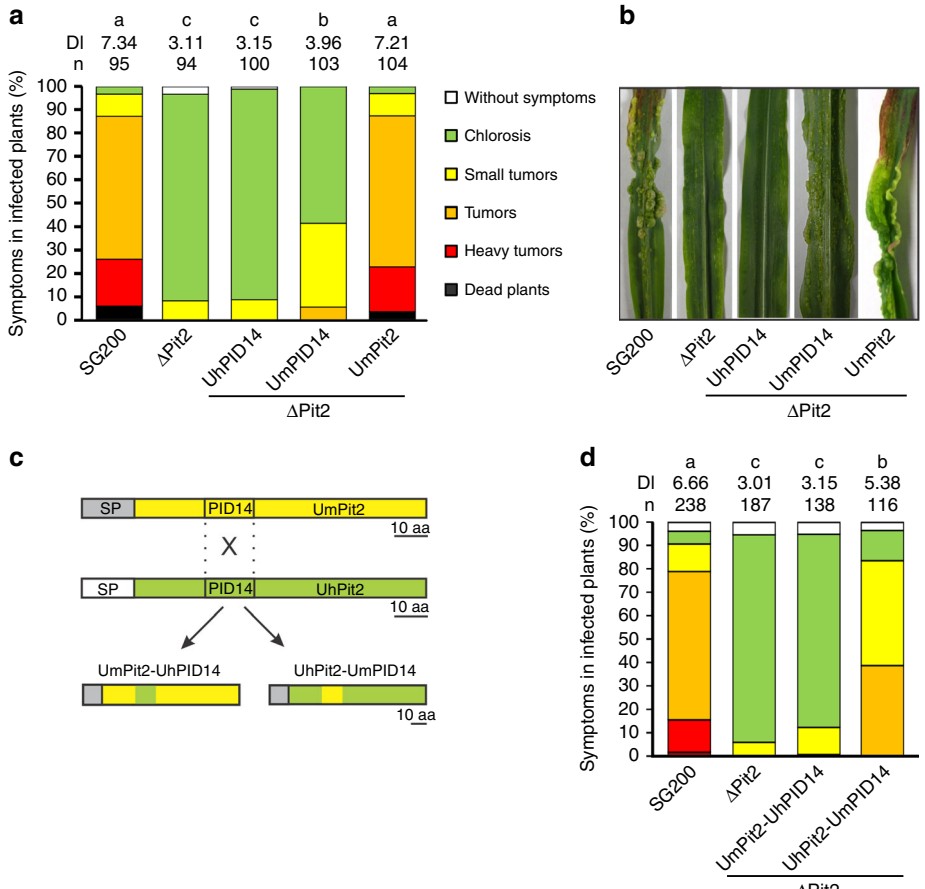

**Fig. 2** UmPID14 itself is a virulence factor in maize. **a** Disease rating of maize seedlings at 12 dpi infected with *U. maydis* SG200 strain, ΔPit2 mutant and complemented ΔPit2 strains with UmPID14 (ΔPit2-UmPID14), UhPID14 (ΔPit2-UhPID14) and UmPit2 (ΔPit2-UmPit2). Number of samples (*n*). DI: disease index calculated as a mean of at least three biological replicates. Letters above indicate significant differences between samples ($\alpha = 0.05$, Tukey test). **b** Disease symptoms of maize seedlings at 12 dpi infected with strains used in (**a**). **c** Strategy to generate chimeras for expression in *U. maydis* SG200 from UmPit2 (yellow) and UhPit2 (green). The signal peptide (SP) from UmPit2 (dark gray) was maintained and the PID14 motif has been exchanged. Gene expression is under control of UmPit2 promotor. **d** Disease rating of maize seedlings 12 dpi with *U. maydis* SG200 strain, ΔPit2 mutant and complemented ΔPit2 strain with chimeras UmPit2-UhPID14 (ΔPit2- UmPit2-UhPID14) and UhPit2-UmPID14 (ΔPit2- UhPit2-UmPID14). DI: disease index calculated as a mean of at least three biological replicates. Letters above indicate significant differences between samples ($\alpha = 0.05$, Tukey test)

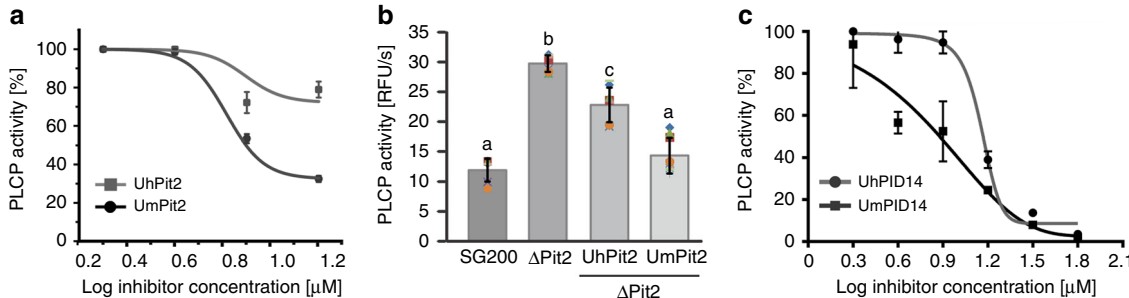

**Fig. 3** Inhibition of maize PLCPs by *U. maydis* and *U. hordei* Pit2/PID14. The activity of PLCPs in apoplastic fluids of maize leaves treated with salicylic acid (SA) was determined using the fluorogenic substrate Z-Phe-Arg-AMC. **a** Concentration range of inhibition using heterologous expressed UmPit2 and UhPit2 proteins. Activity was set to 100% in the absence of inhibitor. Curves were generated using a nonlinear fit and error bars are SEM of three independent replicates. **b** Activity of apoplastic PLCPs during infection. Apoplastic fluids of maize seedlings infected with *U. maydis* SG200, ΔPit2 and the complemented ΔPit2-UhPit2 and ΔPit2-UmPit2 were collected 3 dpi and protein quantification was performed. Apoplastic fluids were set to a concentration of 57 μg/ml total protein and PLCP activity was measured using the fluorogenic substrate Z-Phe-Arg-AMC. Values are means of three independent biological replicates with at least three technical replicates each (dots) and the error bars represent the standard deviation (SD). Letters above error bars indicate significant differences between samples ($\alpha = 0.05$, Tukey test). **c** Concentration range of UmPID14 and UhPID14 peptides was performed. Activity was set to 100% in the absence of inhibitor. Curves were generated using a nonlinear fit and error bars are SEM of three independent replicates

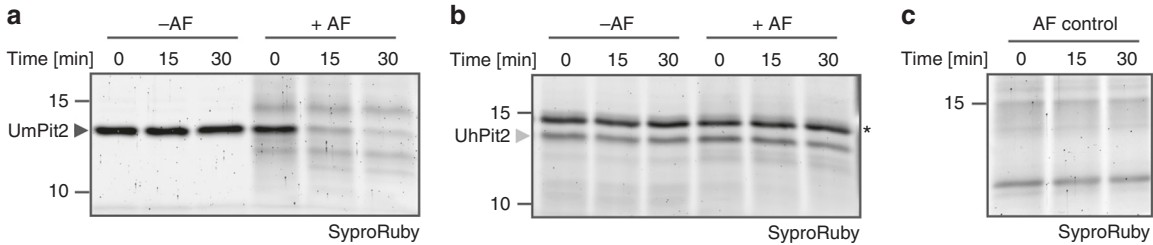

**Fig. 4** UmPit2 is processed in maize apoplastic fluids. Heterologous expressed UmPit2 (**a**) and UhPit2 (**b**) were incubated without (-AF) or with (-AF) maize apoplastic fluids for 0, 15 and 30 min. Reactions were stopped by adding 1× gel loading buffer and samples were analyzed by gel electrophoresis and SyproRuby staining. Asterisk represents a contaminant signal from *E. coli*. **c** Apoplastic fluid control over the time of the experiment

**Fig. 5** PLCPs cleave UmPit2 in the apoplast. **a** Input used to perform inhibitory test of Pit2 stability. Heterologous expressed UmPit2 and maize AF at different time points during the experiment. **b** Addition of E-64 stabilizes Pit2 degradation. Four inhibitors for main classes of proteases: E-64 for cysteine proteases, DCI for serine hydrolases, EDTA for metalloproteases and Pepstatin A for aspartic proteases were preincubated with maize AF for 10 min. Pit2 degradation was analyzed over time (0, 15 and 30 min) using SyproRuby staining. **c** Inhibition of cysteine proteases stabilizes Pit2 in the apoplast. Maize AF was coincubated with a mix of inhibitors containing DCI [10 μM], EDTA [1 mM] and Pepstatin A [10 μM] (Inh. Mix) with or without E-64 [10 μM]. As a control the noninhibitor sample containing DMSO is shown. **d** Fractionation of SA-treated apoplastic fluid by anion-exchange chromatography. Top panel shows elution conditions and the profile of eluted proteins using a salt gradient. Low panel shows PLCPs activity measurements of fractions with E-64 or DMSO as a control using the fluoro-substrate Z-Phe-Arg-AMC. Fraction 24 (F-24, red labeled) showed the highest PLCP activity and was then used for further experiments. **e** PLCPs enrichment of F-24 cleaves Pit2. F-24 was preincubated with 10 μM E-64 or DMSO followed by addition of purified UmPit2. Samples were incubated for 0, 15 and 30 min and analyzed by gel electrophoresis using SyproRuby. **f** UmPit2 is processed at similar rates as an endogenous maize substrate. UmPit2 as well as the endogenous maize PLCP substrate Prozip11 were heterologous expressed in *E-coli*. 3.2 μM proteins were coincubated with F-24 pretreated with inhibitor mix (Inh. mix) in the presence or absence of E-64. Processing of Prozip11 and UmPit2 has been monitored over time. Samples were analyzed by gel electrophoresis and stained with SyproRuby for visualization. As a control purified proteins as well as F-24 were loaded on gel (right gel). DMSO dimethyl sulfoxide, DCI 3,4-dichloroisocoumarin

Fractionation of apoplastic fluids from SA-treated maize leaves by anion-exchange chromatography has been previously shown to successfully enrich immune-activated PLCPs[12,16]. Collected fractions were tested for PLCP activity using the fluorogenic substrate Z-Phe-Arg-AMC. The highest activity was observed with fraction 24 (F-24), whereas fractions inhibited with E-64 did not show any activity (Fig. 5d). Mass spectrometry analysis of F-24 reveals high abundance of the maize PLCPs CP1a, CP1b and CP2 (Supplementary Data 1). These proteases were previously described to undergo activation in the apoplast upon SA-treatment[16], to interact with UmPit2[12] and to be involved in the release of the endogenous immune signaling peptide Zip1 from the propeptide Prozip1[15]. Besides PLCPs, other proteases such as serine hydrolases were found in this fraction (Supplementary Data 1).

To confirm that the PLCPs present in F-24 cleave UmPit2, its stability was followed over time using a preincubation of the fraction with 50 µM E-64 or dimethyl sulfoxide (DMSO) as a control. UmPit2 showed processing over time in the F-24 preincubated with DMSO, but it was stabilized in the fraction preincubated with E-64, providing further evidence for PLCPs-driven processing of UmPit2 (Fig. 5e). Similarly, incubation of UmPit2 with papain, the type member of the PLCPs, resulted in processing of UmPit2 over time, confirming UmPit2 as a substrate for PLCPs (Supplementary Fig. 3). Those experiments indicate that UmPit2 is a suitable substrate for host PLCPs.

To address the question whether UmPit2 can be a substrate similar to maize endogenous PLCP substrates, we performed a competition experiment using the heterologous expressed maize propeptide Prozip1. Processing of Prozip1 by the PLCPs CP1 and CP2 generates Zip1, which activates SA-associated defense responses in maize[15]. F-24 pretreated with an inhibitor mix in the absence of E-64 was incubated with same molarity of Prozip1 and UmPit2 and processing was monitored over time. Both Prozip1 and UmPit2 are cleaved at similarly same rates (Fig. 5f) while addition of E-64 results in stabilization of Prozip1 and UmPit2 (Fig. 5f). This experiment indicates that UmPit2 is processed by PLCPs with similar efficiency as the endogenous Prozip1 propeptide.

For a more detailed characterization of the PLCP-mediated processing of UmPit2, mass spectrometry (MS) analysis of F-24 incubated with UmPit2 was performed. To find UmPit2 peptides specifically produced by PLCP cleavage, F-24 was pretreated with a mix of inhibitors for metalloproteases, aspartic proteases and serine proteases but not E-64. As a control, UmPit2 was incubated with F-24 pretreated with only E-64 to identify cleavage of non-PLCP proteases. After incubation, samples were precipitated using TCA and small peptides remaining in the soluble fraction were analyzed. An average of three MS runs of independent biological replicates including only peptides present in at least two replicates resulted into a total of 28 peptides exclusively identified in the sample pretreated with the inhibitor mix and without E-64, corresponding to a mean of 57.7% total protein coverage, and representing peptides cleaved by PLCPs. Remarkably, no peptides were exclusively identified in the sample preincubated with E-64 and only four peptides were identified as shared peptides present in both samples (Fig. 6a; see Methods for details). The peptide sequence RRWWFG could not be identified in any of the MS runs, even after the use of 3 M guanidine, forming a gap in the most important residues previously identified to be necessary for inhibition[12]. Therefore, we considered those six residues as the "putative docking site", which likely serve as anchor residues to facilitate inhibition (Fig. 6a, orange line). Notably, the majority of UmPit2 peptide products found by mass spectrometry showed a preference for hydrophobic amino acids at the P2 position such as Val, Leu, Ile

and Phe (Fig. 6a, blue letters and dashed lines), correlated with previous studies where PLCPs mostly show substrate specificity for hydrophobic amino acids at the P2 position[25–28]. To independently determine the dominating activity in F-24 and confirm the P2 specificity for maize PLCPs, a Proteomic Identification of protease Cleavage Sites (PICS) analysis of F-24 was performed[29,30]. Proteome-derived peptide libraries prepared by the digestion of E. coli cell lysates with trypsin or GluC were incubated with F-24 and cleaved peptides were identified by quantitative mass spectrometry. Cleaved semi-specific peptides were used for reconstruction of the cleavage sites and positional amino acid occurrence visualized by iceLogos[31]. F-24 shows at the P2 position substrate specificity for Leu, Pro, and Val (Supplementary Fig. 4). Remarkably, this substrate specificity at P2 position observed by PICS coincides with the cleavage sites observed in UmPit2 after incubation with F-24 (Supplementary Fig. 4 and Fig. 6a). While PLCPs typically display a preference for Lys and Arg at the P1 position[25], no such preference was observed for F-24 at this position, probably due to the presence of carboxipeptidases in the F-24 sample (Supplementary Data 1). Together, this evidence confirms that UmPit2 is a suitable substrate of maize PLCPs and undergoes processing in the apoplast.

**PLCPs release an inhibitory portion embedded in UmPID14.** UmPit2 peptides recovered and found in the supernatant after TCA precipitation belong mostly to the UmPID14 and neighboring region, indicating that the processing occurs in close proximity of the inhibitory portion and inside of the PID14 region. To confirm that UmPID14 itself can be cleaved by PLCPs, F-24 was preincubated with an inhibitor mix in the presence or absence of E-64 followed by 30-min incubation with UmPID14. Subsequently, peptides in the samples were analyzed by mass spectrometry. Only peptides present in at least two of the three replicates were taken into account. Cleavage of UmPID14 was found in both samples. Interestingly, only samples preincubated with EDTA, pepstatin A and DCI but not E-64 contained the specific peptides KLNRRWW corresponding to the N-terminus of UmPID14 and WWFGFTGSL corresponding to the C-terminus, confirming processing of UmPID14 by PLCPs (Supplementary Fig. 5). A substrate competition experiment demonstrates that UmPID14 can be a substrate, but UmPit2 is preferentially cleaved rather than UmPID14 by the apoplastic PLCPs (Supplementary Fig. 6).

Next, we hypothesized that in order to facilitate an interaction with PLCPs, the inhibitory UmPID14 motif should be surface localized. To address this assumption a structural-model analysis has been performed for UmPit2 without signal peptide using the iTASSER software tool (http://zhanglab.ccmb.med.umich.edu/I-TASSER/). The best model predicted (C-score = −3.35) suggests that most of the amino acids of the UmPID14 motif are surface exposed, making them available for interaction with other molecules (Fig. 6b). Interestingly, amino acids K44 and R47 form a free hook-like structure that might facilitate contact with other molecules (Fig. 6c). We used the model to predict the position of the putative docking site, RRWWFG. The hydrophobic bulky tryptophan residues as well as F51, the putative P2 cleavage position at the C-termini, are inside the molecule whereas the flanking P2 cleavage position L45 at the N-termini of the docking site appeared to be surface localized (Fig. 6d). The localization of the docking site as well as its flanking P2 recognition sites by PLCPs suggest that in order to release the inhibitory portion first cleavage of UmPit2 should take place.

The findings that UmPit2 is processed by PLCPs and that although the majority of UmPID14 residues are surface localized

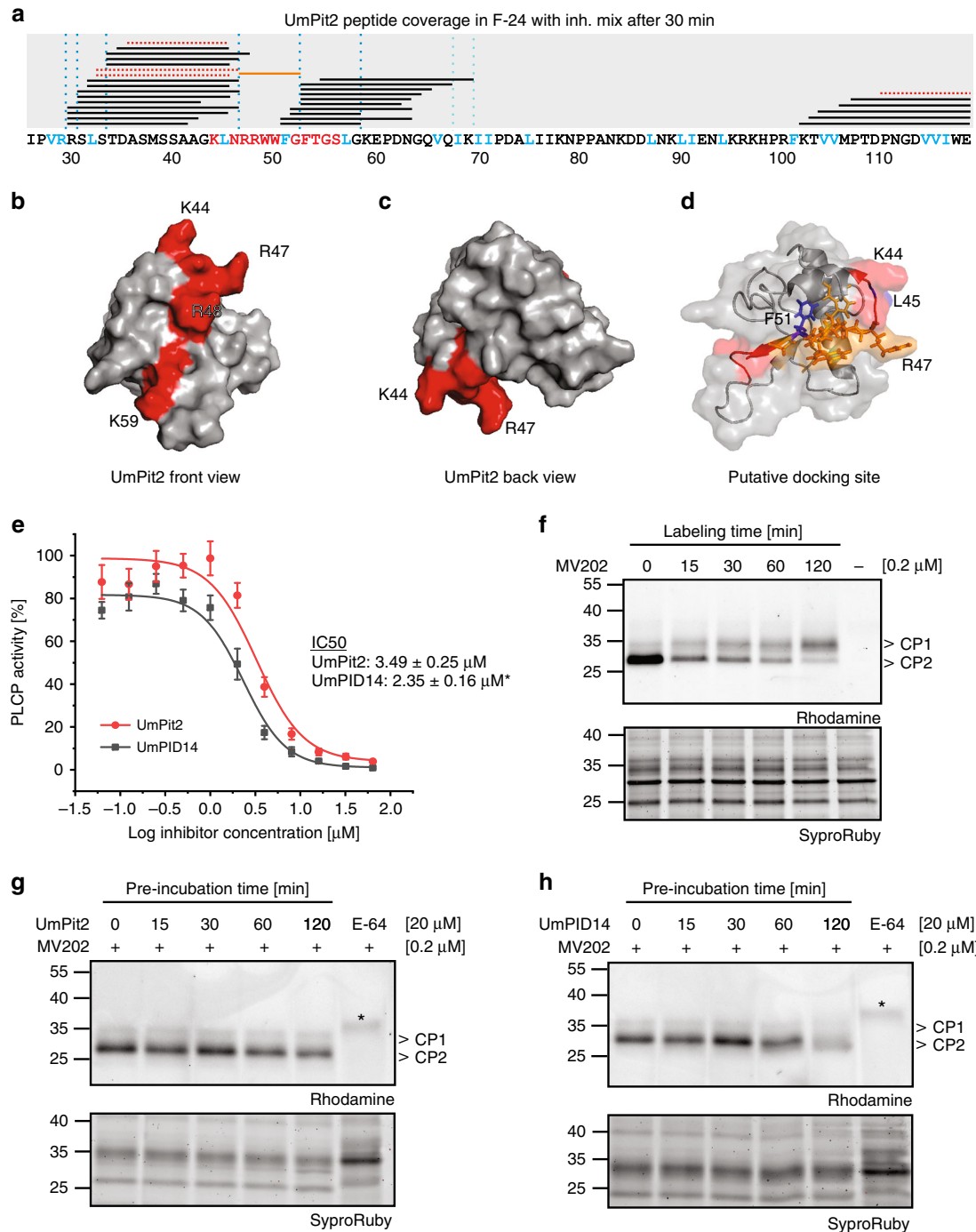

except the docking site suggest that the inhibitory portion contained in the PID14 motif should be released to better perform inhibition. To address this question, an inhibitory profile was performed comparing the affinity of UmPit2 and UmPID14 to maize PLCPs. Activity of PLCPs in apoplastic fluids was monitored using the substrate Z-Phe-Arg-AMC. AF activity without inhibitor was set to 100% and the inhibitory profiles of UmPit2 and UmPID14 were monitored in a concentration range manner. This experiment showed that UmPID14 is significantly more effective in inhibiting PLCPs than the full UmPit2 protein (Fig. 6e) and suggests that both cleavage of PLCPs and their further inhibition are required for Pit2 virulence function. Notably, stability experiments of the chimeras UmPit2-UhPID14 and UhPit2-UmPID14, as well as PLCP activity assays during maize infection reveal that UmPit2-UhPID14 is gradually

processed when incubated with apoplastic fluids but fails to inhibit PLCP activity in planta (Supplementary Fig. 7). Conversely, the chimera UhPit2-UmPID14 is stable in apoplastic fluids but can partially suppress PLCP activity in leaves at 3 dpi (Supplementary Fig. 7). These experiments are in line with the in planta phenotypes observed for the fungal mutants expressing Pit2-chimeras. Here, expression of UhPit2-UmPID-14 partially complements tumor formation, whereas UmPit2-UhPID14 fails to restore virulence (Fig. 2d). Taken together, these experiments show that cleavage of Pit2 is not essential for the inhibitory activity in vitro but it is necessary and required for full virulence function.

To monitor the activity of maize PLCPs present in the apoplastic fluids, activity-based protein profiling was performed using the probe MV202. This probe is a DCG-04 derivative that

**Fig. 6** UmPit2 cleavage releases the inhibitory portion. **a** MS analysis of UmPit2 with F-24. 30 min incubation of UmPit2 with F-24 in the presence of inhibitor mix without E-64 was performed and reaction was stopped by adding TCA. Only peptides released during incubation and remaining in the supernatant were analyzed. The UmPit2 sequence without signal peptide including the PID14 motif (red amino acids) is shown. Black lines above the sequence represent the peptides found in the analysis. Red lines represent peptides found in the sample preincubated with E-64. The orange line represents a peptide that was reproducibly not identified in the MS analysis and might serve as a docking site. Putative cleavage sites based on peptide coverage after PLCP recognition of Val, Arg, Leu; Ile or Phe at the P2 position (blue amino acids) are shown as blue dotted lines. This figure shows the average of three MS runs of three biological replicates. **b–d** Surface modeling of UmPit2 without signal peptide based on ITASSER prediction analysis (http://zhanglab.ccmb.med.umich.edu/I-TASSER). The PID14 region is labeled in red. The majority of UmPID14 residues are surface exposed (front view, **b**). K44 and R47 form a distinctive hook-like structure (back view, **c**). L45 is surface localized whereas F51 resides inside UmPit2. The putative docking site with the sequence RRWWFG is shown in orange sticks and the majority of residues are surfaced localized. **e** UmPID14 inhibits PLCPs more efficiently than UmPit2. The activity of PLCPs in apoplastic fluids of maize leaves treated with salicylic acid (SA) was determined using the fluorogenic substrate Z-Phe-Arg-AMC. Inhibitory profiles of UmPID14 (black) and UmPit2 (red) in a concentration range against PLCP activity were compared. Error bars show the standard error of the mean calculated for three independent biological replicates. $IC_{50}$ values were calculated as the mean of three biological replicates using a nonlinear curve fit. $IC_{50}$ values are statistically significant at the 0.05 level ($p = 0.0167$) represented by an asterisk. **f** MV202 time-course labeling of SA-treated apoplastic fluids (AF). Labeling of maize AFs was performed using 0.2 μM MV202 in a one-pot reaction and samples were collected at different time points. The main two signals observed correspond mainly to CP1 and CP2, previously characterized by ref. [21]. Samples were analyzed by gel electrophoresis and monitored using a rhodamine filter. After analysis samples were stained with SyproRuby and used as loading control. **g** UmPit2 competition experiment. SA-treated maize AFs were preincubated in a one-pot reaction with 20 μM UmPit2 and at different time points were collected and then labeled with 0.2 μM MV202 for 1 h. E-64 was used as a control. Asterisk represents a background signal. Samples were analyzed as described in (**f**). **h** UmPID14 competition experiment. SA-treated maize AFs were preincubated at different time points with 20 μM UmPID14 and then labeled for 1 h with 0.2 μM MV202. E-64 was used as a control and asterisk represents a background signal. Samples were analyzed as described in (**f**)

contains the E-64 warhead which binds covalently and irreversible to the active site of PLCPs. MV202 contains two affinity tags, a bodipy fluorescent group and a biotin affinity tag[28]. A time-course experiment of SA-treated maize apoplastic fluid labeled with MV202 was performed to monitor transient changes of the PLCP activity. In accordance to the previous identification using a DCG-04 pull-down followed by MS analysis[21], two main signals at ca. 30 KDa can be observed (Fig. 6f). Based on the previous identification the upper band corresponds mainly to CP1a and CP1b, whereas the lower band corresponds mostly to CP2 (Fig. 6f). MV202 rapidly labels CP2, since immediately after addition of the probe, at time point 0 min, there is an intense band corresponding to CP2, whereas CP1 is weakly labeled. This labeling pattern can be attributed to a higher affinity of the probe to CP2, to higher abundance of CP2 in comparison to CP1 or to a less active CP1 at the start point of the experiment. Over longer incubation time with the probe, the activity of CP1 continuously increases, whereas the signal for CP2 decreases suggesting that activation of CP1 leads to a direct or indirect degradation of CP2 (Fig. 6f). This experiment indicates that apoplastic PLCP activity is a dynamic process. Inhibition of apoplastic fluids, caused by the addition of the probe, results in a coordinate mechanism of activation and degradation of CP1 and CP2 proteases. To test if UmPit2 and UmPID14 can affect this dynamic process of degradation and activation of PLCPs, a competition experiment was performed. SA-treated apoplastic fluids were preincubated over time with either UmPit2 or UmPID14, respectively, followed by 60-min labeling with MV202. At this labeling time point, CP1 and CP2 show similar intensities in the absence of UmPit2/UmPID14 (Fig. 6f). In the presence of UmPit2 the CP2 signal is stabilized over time, while signals for CP1 remained faint throughout the time-course. In addition, the overall activity of both CP1 and CP2 was decreasing over time, likely reflecting PLCP-inhibition by UmPit2 (Fig. 6g). These results indicate that UmPit2 interferes with the apoplastic PLCP dynamics and particularly might prevent the activation of CP1. A similar finding was made for UmPID14 (Fig. 6h), where especially during the first 30 min preincubation with UmPID14, stabilization of CP2 but faint signals for CP1 were observed (Fig. 6h). In contrast to UmPit2, longer incubations with UmPID14 resulted in an almost complete suppression of specific signals, including CP2, indicating full PLCP-inhibition. In line with the concentration-

dependent inhibition profile (Fig. 6e), this finding confirms that UmPID14 is a more effective inhibitor of PLCPs compared to the full-length UmPit2 effector. These experiments imply that UmPit2 might target CP1 first, likely by having a higher binding capacity as a substrate, to prevent further activation of CP2. To test this hypothesis, inhibitory profiles of *N. benthamiana*-produced CP1 and CP2 were performed. UmPit2 inhibitory curves suggest a higher inhibitory activity for CP2 than for CP1, although in both cases full inhibition was not reached. By contrast, UmPID14 shows significantly stronger inhibition of CP1 (Supplementary Fig. 8A-B).

Altogether the present findings demonstrate that UmPit2 virulence function depends on two main features: its capacity to be a suitable PLCP substrate, as well as its inhibitory function contained in the UmPID14 motif.

**UmPID14 is conserved among plant associated microbes.** The essential function of Pit2 for *U. maydis* virulence together with the crucial role of apoplastic PLCPs for plant defense signaling raises the question, if Pit2-like proteins can also be found in other organisms unrelated to smut fungi. To this end, blastp search analysis was conducted using UmPit2 as a template in all downloadable proteome databases from GenBank (www.ncbi.nlm.nih.gov/genbank/). Six Pit2-like sequences were found mainly in smut fungi (four sequences) and related yeast-like fungi (two sequences) from the Ustilaginaceae family. Surprisingly, five sequences were found in bacteria, of which four were Gram-positive species and one (*Burkholderia vietnamiensis*), a Gram-negative proteobacterium. Phylogenetic analysis of the 11 Pit2-like sequences inferred using the minimum evolution method resulted in mainly two clusters: one for bacteria and one for fungi (Fig. 7a). Based on the bootstrap values Pit2-like bacterial sequences shared higher sequence similarities among them than fungi protein sequences (Fig. 7a). Interestingly, the most conserved sequence found using a motif search with the 11 Pit2-like proteins matches to the PID14 region (Fig. 7b). Extracted sequences that are part of the motif were used to generate an iceLogo that represents the most conserved and significant amino acids (E-value: 2.1e-0.34) found in bacteria, yeast-like and smut fungi (Fig. 7c). Remarkably, this motif reported by the motif-based sequence analysis tool, MEME[32], is very similar to

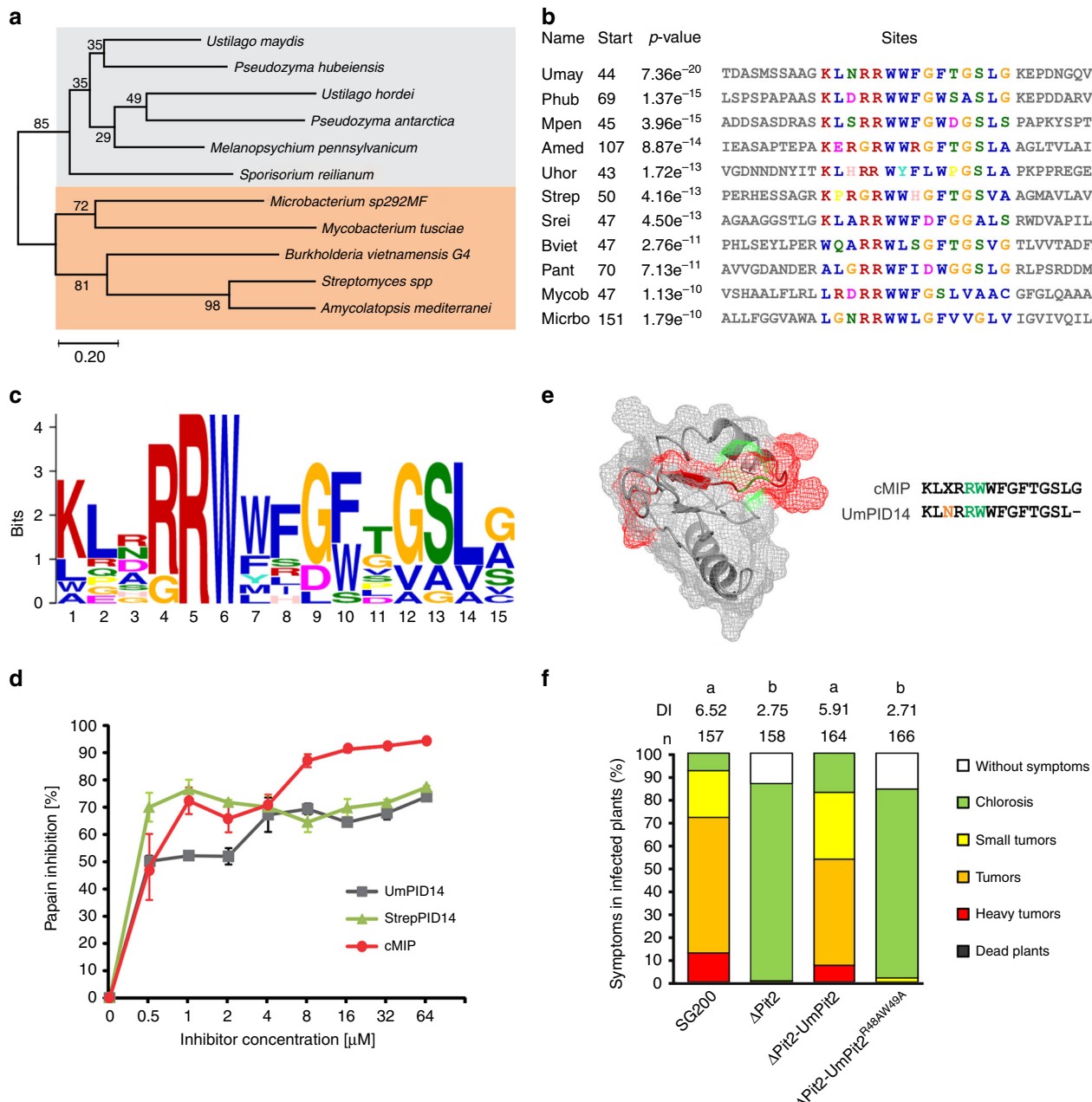

**Fig. 7** PID14 is conserved in bacteria and fungi and its function depends on two amino acids. **a** Pit2 evolutionary relationships of sequences found in bacteria and fungi. Using blastp and the Pit2 sequence from *U. maydis* as template, 11 homologous sequences were found in bacteria and fungi. Their evolutionary history was inferred using the Minimum Evolution method. The optimal tree with the sum of branch length = 6.46244084 is shown. The percentage of replicate trees in which the associated taxa clustered together in the bootstrap test (2000 replicates) is shown next to the branches. **b** PID14 is the most conserved motif in Pit2 sequences. The 11 Pit2 sequences were used for a search of sequence motifs using MEME. Extracted sequences that are part of the motif are shown. **c** Conserved microbial inhibitor of proteases (cMIP). Icelogo representing the most conserved amino acids made of 11 sequences found in bacteria, yeast and smut fungi containing the cMIP motif using MEME. cMIP appears to be similar to UmPID14 except there is one more amino acid at the end of the motif. **d** Inhibitory profile of bacterial and fungi cMIPs against papain. The activity of papain was determined using the fluorogenic substrate Z-Phe-Arg-AMC. Values without inhibitor were set to 100% activity. UmPID14 (gray), StrepPID14 (green) and cMIP (orange) inhibition of papain was monitored in a concentration range between 0 and 64 μM. Log inhibitor concentrations were plotted against papain activity. Error bars represent the standard deviation of three replicates. Shown is a representative plot of five independent replicates showing similar results. **e** UmPit2 structural model representing the PID14 motif (red) and the localization of the most conserved amino acids RW (green) found in cMIP. UmPID14 and cMIP sequence comparison. In red dissimilar amino acids and in green the conserved RW. **e** UmPit2 structural model representing the PID14 motif (red) and the localization of the most conserved amino acids RW (green) found in MEME. **f** Disease rating of maize seedlings at 12 dpi infected with *U. maydis* SG200 strain, ΔPit2 mutant and complemented ΔPit2 strains with UmPit2 (ΔPit2-UmPit2) and the mutant strain ΔPit2-UmPit2^R48AW49A, carrying R48A and W49A point mutations

UmPID14 except that there is one more C-terminal amino acid. Based on this analysis, we propose that the previously *U. maydis*-specific PID14 motif might represent a conserved microbial inhibitor of proteases (cMIP). To confirm that the inhibitory function of PLCPs is conserved, a bacterial cMIP (StrepPID14, 14 amino acids) and a fungi cMIP (UmPID14, 14 amino acids), as well as the conserved motif found by MEME (cMIP, 15 amino acids) were tested for their ability to inhibit papain. Activity of papain was monitored over time using the substrate Z-Phe-Arg-AMC and the inhibitory efficiency of UmPID14, StrepPID14 and cMIP was tested in a concentration range. Inhibition of papain was calculated based on the sample without inhibitor. All tested cMIP peptides showed a concentration-dependent inhibition of papain, confirming that bacterial and fungi cMIPs are functional inhibitors of PLCPs (Fig. 7d). Interestingly, the cMIP motif contains the two highly conserved amino acids R5 and W6 (Fig. 7c). The protein model of UmPit2 was used to reveal the localization of these two main residues of cMIP. Residues R5 and W6 are nonexposed located at the base of the hook-like structure (Fig. 7e). To confirm if the RW residues are important for UmPit2 function, a *U. maydis* strain carrying Pit2$^{R48AW49A}$ mutations was generated and tested for their virulence in planta. Comparison of disease rating at 12 dpi between SG200 and the complemented ΔPit2_Pit2 strain did not show differences in virulence (Fig. 7f). In contrast, the ΔPit2_Pit2$^{R48AW49A}$ strain cannot complement tumor formation resembled the avirulent phenotype of the ΔPit2 strain (Fig. 7f). This experiment confirms that the conserved RW residues are essential for Pit2 virulent function in planta.

Altogether we have identified cMIP, a functional inter-kingdom motif found in bacteria and fungi capable of inhibiting PLCPs.

## Discussion

This study shows the mechanism of action of the *U. maydis* effector protein Pit2, a molecular mimicry substrate and an inhibitor of plant PLCPs which has been described to be conserved in different smuts. We found that UmPit2 is processed by PLCPs in apoplastic fluids of maize leaves thereby releasing the inhibitory portion necessary for PLCP inhibition. UmPit2 virulence function requires processing by PLCPs, which is followed by their inhibition. While Pit2 has been classified as a core effector of smut fungi[3], its mechanism of action might not be conserved across smut pathogens, suggesting functional diversification of this effector. Recently, functional diversification between *U. maydis* and *S. reilianum* was demonstrated for the effector Tin2[33]. This neofunctionalization of Tin2 is thought to correlate with the differential infection styles of the two pathogens[33]. It is remarkable that UhPit2 cannot complement virulence function in maize, despite it contains the conserved inhibitory motif PID14, as well as *S. reilianum* Pit2 can only partially complement *U. maydis* virulence. This lack in functional conservation of Pit2-orthologs might be indicative for a neofunctionalization of this effector. In line, the PID14 motif of *U. hordei* has a reduced inhibitory activity on maize PLCPs compared to UmPID14 which again might reflect an adaptation to the different host plants. Interestingly, we observed that the SA-induced induction of apoplastic PLCPs is not found in barley (Supplementary Fig. 9). Unlike the situation in maize, this indicates that SA-dependent immune response in barley does not involve apoplastic PLCPs. In accordance, UhPit2 is not processed in barley apoplastic fluids treated with or without salicylic acid (Supplementary Fig. 10). Altogether, our data indicate a conserved function of the PID14 motif, while the effector itself might be subject of functional diversification in a host-specific manner.

After cleavage of its signal peptide, the mature Pit2 protein does not contain any cysteine residues. This is unusual for an apoplastic effector, since cysteine residues are thought to maintain protein integrity and stability in the apoplast and are hallmarks when searching for putative secreted effectors[34]. Consequently, the Cys-free UmPit2 is rapidly degraded when incubated with maize apoplastic fluids. This situation may appear counter-intuitive, since the canonical expectation would be that the effector should be stable in the apoplast to fulfill its function as a PLCP inhibitor.

Pit2 orthologs from different smuts failed to complement tumor formation in maize, while the UmPID14 itself acts as a virulence factor in *U. maydis*. However, virulence complementation by UmPID14 remains incomplete compared to the full-length Pit2 effector. An explanation for this might be that not only inhibition of PLCPs but also the recognition as a substrate is required for full virulence. The substrate preference of PLCPs supports this hypothesis since UmPit2 is favored as a substrate rather than UmPID14 indicating that UmPID14 is not "sequestering" PLCPs as effectively as UmPit2. Moreover, hints for an importance of the PID14-surrounding regions arise from the finding that sequences from two different *Sporisorium reilianum* formae speciales showed signs of positive selection both inside and outside the conserved PID14 motif[35]. Interestingly, the chimeric protein UmPit2_UhPID14 processed in the apoplast but cannot rescue tumor formation although in vitro experiments showed that UhPID14 has the potential to inhibit maize PLCPs. However, UhPit2 is a rather weak inhibitor of maize PLCPs in vitro and during infection, indicating that there is a threshold of PLCPs inhibition that needs to be passed in order to allow tumor formation. These findings point towards a host-specific dual function of UmPit2: on one side it might attract PLCPs as a substrate to prevent additional posttranslational activation of PLCPs or other signaling components and on the other side after UmPit2 cleavage, released peptides could more efficiently perform inhibition.

Pit2 shows a very high sequence divergence among smut fungi. This sequence divergence is a hallmark of a high evolutionary pressure on Pit2. Remarkably, the majority of positive selected sites that were identified in Pit2 of *S. reilianum* f. sp. zeae are next to Leu, Val, Arg and Phe residues[35] which are the preferred P2 positions of PLCPs in the maize apoplast and confirmed by our PICS analysis (Fig. 6a and Supplementary Fig. 4). This finding is also in accordance to the literature where the specificity requirements at P2 position for bromelain, papain and several other plant PLCPs have been similarly described[25–27]. In line with this, UmPit2 is also processed by papain, confirming that UmPit2 is a substrate for PLCPs in general. Apoplastic PLCPs are well known as key players during plant immunity[13] and are mostly posttranslational activated either by self-cleavage of the propeptide or by cleavage of the propeptide by another PLCP[36]. Besides, spacial and temporal regulation, such as low pH, serves as determinants of proteolysis in the cell[37]. PLCPs might work as sensors of unknown or high accumulating molecules in the apoplast, degrading them and generating peptides for further activation of signaling cascades. Once conditions are maintained and active PLCPs present in the apoplast, acting as sensors of self and non-self-material, are "busy" cleaving substrates most likely self-activation of other PLCPs will not occur. Processing of UmPit2 by PLCPs might release the inhibitory portion as well as other peptides nearby. We could not identify the full sequence of UmPID14 using MS analysis of UmPit2 incubated with F-24 (Fig. 6a). This could indicate that UmPID14 is further processed and it contains the inhibitory portion that stays bound to the PLCPs and therefore not free for MS identification. Indeed, MS analysis of UmPID14 incubated with F-24 confirms this further

processing. UmPID14 seems to be more susceptible for cleavage by other proteases than UmPit2 since we found two peptides exclusively present in the sample preincubated with E-64, likely products of other proteases but not PLCPs present in the fraction (Supplementary Fig. 5).

The maize PLCPs CP1 and CP2, which are also main inter-actors of Pit2, are responsible for the release of the endogenous signaling peptide Zip1. Zip1 triggers an SA defense response and, in a positive feedback loop, further activates the apoplastic PLCP activity[15]. In this line, one can hypothesize that blocking of CP1 and CP2 by UmPit2 is a fungal strategy to prevent activation of plant PLCP-dependent immune signaling. Efficient binding, and sequestering, of the immune-relevant PLCPs in the apoplast by UmPit2 is mediated by several preferred P2 binding sites sur-rounding the PID14 region, which is surface-exposed and parti-cularly shows a higher inhibitory affinity for PLCPs in vitro. We suggest that after processing in the close proximity to its target, the inhibitory portion within the UmPID14 motif remains bound to the PLCP, resulting in inhibition. Together, this can be sum-marized in a model, where the UmPit2 effector acts as a mole-cular mimicry substrate, which first "attracts" the PLCPs by being a suitable substrate. Processing of UmPit2 by PLCPs will release the inhibitory portion from the effector, which prevents further activation of the PLCP-mediated immune response (Fig. 8). Evolutionary arms race shaped Pit2 into a highly diversified core effector that harbors a conserved inhibitor peptide embedded within a host-specific surface structure.

Given the high degree of diversification of Pit2 among smut fungi, it was surprising to find conservation of the PID14 region in unrelated species, including bacteria. A motif search using the bacterial and fungi Pit2 sequences identified 15 amino acids that resembled PID14 including one additional residue at the C-terminus. Remarkably, PID14 motifs from bacterial and fungi, as well as the motif sequence found by MEME are efficient inhibi-tors of papain confirming their PLCP inhibitory function.

Because this functional inter-kingdom motif resembles PID14, we renamed it into the "conserved microbial inhibitor of proteases" (cMIP). Mutation of the two highly conserved residues R5 and W6 in the PID14 region of UmPit2 resulted in a complete loss of U. maydis virulence demonstrating their essential function in virulence. These data are in line with our previous report, where mutations of the UmPID14 residues WWFGF into GGAGG resulted in both loss of PLCP-inhibition and virulence[12]. How-ever, it still need to be analyzed if mutations in the RW residues of cMIP also result in loss of inhibitory function.

Identified microbes containing cMIP are not only pathogens, but some have been described as plant endophytes with beneficial impact on their hosts. We therefore hypothesize that inhibition of PLCPs by cMIP-containing proteins could be an ancient micro-bial strategy to suppress plant immune responses to establish and maintain a compatible interaction. Although the exact mechan-ism how endophytes suppress immune responses has not been yet identified, there are some indications that bacterial endophyte strains can evade plant defense[38,39]. None of the bacterial sequences that contain the cMIP motif are predicted to be secreted (Supplementary Data 2). The majority of cMIP-containing sequences are annotated as either hypothetical or unknown proteins with the exception of the sequence from the Gram-positive Microbacterium that was annotated as a tripartite tricarboxylate transporter (TTT) TctB family protein. In contrast, all pathogenic smut fungi were predicted to contain a signal peptide (Supplementary Data 2). Public sequences for the Usti-laginales yeast Pseudozyma antartica and Pseudozyma hubeiensis were not predicted to contain a signal peptide (Supplementary Data 2); however, a manual search for Pit2 homologous sequences predicts an open reading frame containing a secretion signal for P. hubeiensis (Supplementary Fig. 11). With respect to a potential evolutionary origin of cMIP, it is interesting that the cMIP motif found in Burkholderia vietnamiensis is annotated as part of a creatinase N-terminal domain in Xaa-Pro-peptidases

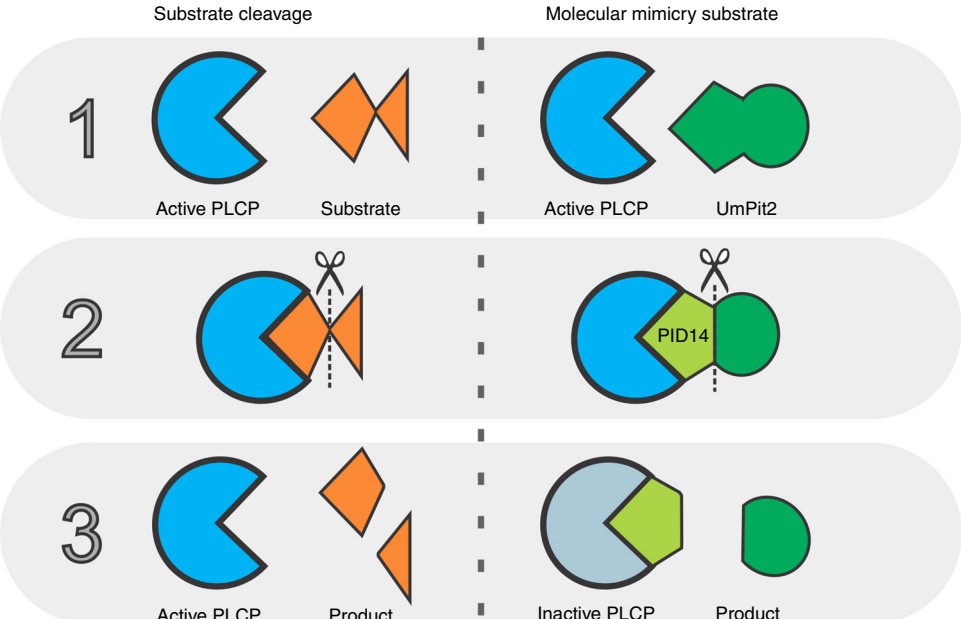

**Fig. 8** Pit2 effector act as molecular mimicry. Mechanism of substrate cleavage by active PLCPs (left panel) consisting of substrate recognition (1), substrate cleavage (2) and product release (3). To avoid activation of the immune system, U. maydis secretes the apoplastic effector Pit2 that contains the PID motif of 14 amino acids. In the molecular mimicry substrate model (right panel), active PLCPs recognize Pit2 as a substrate (1). Next, PLCPs cleave Pit2 similar to endogenous substrates (2). While cleavage products endogenous substrates are released from the active site, the inhibitory portion of Pit2 embedded in the PID14 remains bound to the PLCP and thereby blocks its activity

with unknown function. Xaa-Pro dipeptidyl peptidases are serine hydrolases of the clan SC with a narrow specificity for proline in the P1 position[14]. Pro-domains are key factors to control post-translational activation of proteases and their main function is to keep proteases in an inactive state but ready to be activated upon stimulus. How cMIP inactivates PLCPs at the molecular level and its structure−function relationship are further questions to be investigated in the near future.

## Methods

**Generation of fungal strains.** *U. maydis* strains used in this study are listed in Supplementary Data 3. The plasmid p123_Ppit2 containing the native promotor of Umpit2 (um01375) was generated by restriction-ligation using *Xma*I sites. p123_Ppit2_Pit2SP containing the signal peptide of *Umpit2* under its native pro-motor was further generated by restriction-ligation of p123_Ppit2 and a PCR product specific for *Umpit2* signal peptide containing *Xma*I and *Xba*I restriction sites. p123_Ppit2_Pit2SP was used for the generation of plasmids containing the *pit2* cDNA sequence of different orthologs without their signal peptide sequence (uh02064, sr10529 and mp4_3204_1) and the PID14 regions from *U. maydis* and *U. hordei* using specific gene primers containing *Xma*I and *Xba*I restriction sites. To originate the UmPit2_RW^mut strain site-directed mutagenesis on p123_Ppit2_Umpit2[11] was performed using the QuikChange Multi Site-Directed Muta-genesis kit according to the instructions of the manufacturer (Agilent Technologies, Santa Clara, USA). Constructs were used to generate the complementation strains though homologous recombination in the *ip*-locus providing carboxin resistance[40]. All generated plasmids were confirmed by sequencing and newly produced *U. maydis* strain mutants by southern blot analysis. Transformation and isolation of genomic DNA of *U. maydis* were performed as described before[41]. All strains used in this study are listed in Supplementary Data 3 and oligonucleotides used for cloning in Supplementary Data 4.

**Plant infections and SA-treatment.** Seven-day-old seedlings of Early Golden Bantam (EGB) maize plants were inoculated with *U. maydis* strains with an OD = 2. Severity of disease symptoms was scored 12 days post infection (dpi) as described before[42]. Disease severity was classified as follows: (0) no symptoms; (1) chlorosis; (2) ligular swellings; (3) small tumors, less than 1 mm in diameter; (4) tumors larger than 1 mm in diameter, not associated with bending of stem; (5) large tumors associated with bending of infected stems. Salicylic acid treatment of EGB-maize seedlings was performed via infiltration of a 2 mM SA (Sigma-Aldrich) solution in 1% ethanol as previously described[16].

**CP1 and CP2 overexpression in *N. benthamiana*.** Maize *cp1a* (GRMZM2G166281) without the sequence encoding for the granulin domain and *cp2* (GRMZM2G038636) genes were amplified by PCR from maize (EGB) cDNA. PCR products were directly cloned with *Bsa*I overhangs into the level 1 binary vector (pICH47732) (Addgene) containing 2x35S promotor and a C-terminal Streptwin tag using the golden gate procedure[43]. Binary plasmids pL1M-F1-CP2-Streptwin::2x35S and pL1M-F1-CP1A_nogran-Streptwin::2x35S were generated and transformed into *Agrobacterium tumefaciens* GV3101 competent cells for overexpression in *N. benthamiana*. GV3101 strains containing the desired plasmids were grown in liquid media overnight and diluted to OD = 2 in 10 mM magnesium chloride with 200 μM acetosyringone final concentration (Sigma-Aldrich, Tauf-kirchen, Germany). After 1 h incubation cultures were mixed with GV3101 strains containing the p19 construct[44] to a final OD = 1 and cultures were syringe infil-trated in leaves of 5−6-week-old *N. benthamiana* plants. Three days post infil-tration leaves were harvested and apoplastic fluid was isolated.

**Apoplastic fluid isolation from maize and *N. benthamiana*.** Infiltrated leaves were harvested and collected in a beaker filled with water. Leaves were vacuum infiltrated using a vacuum pump (Vacuubrand GmbH + Co., Wertheim, Germany) three times for 10 min at 60 mbar with an interval of 2 min atmosphere pressure to remove gases. After vacuum infiltration leaves were transferred to 50 ml syringes hanging in 50 ml falcon tubes and centrifuged for 20 min at 2000 × *g* and 4 °C to extract the apoplastic fluid.

**Constructs and heterologous expression in *E. coli*.** For UmPit2 expression in *E. coli*, the plasmid pRSET-GST-PP-Pit2[12] was used. For UhPit2 expression in *E. coli*, uh02064 was amplified by PCR without its signal peptide from cDNA using the primers UhPit2_Xma_fw and UhPit2_Xba_rv (Supplementary Data 3). The PCR product was digested using *Xho*I and *Eco*RI and ligated into the pRSET-GST-PP-Pit2 to generate pRSET-GST-PP-UhPit2. For chimeras UmPit2-UhPID14 and UmPit2-UhPID14, similar cloning strategy as for UhPit2 was followed and PCR fragments were cloned into pRSET-GST-PP-Pit2. Plasmids were transformed into Tuner (DE3) pLysS competent cells (Novagen/Merk, Darmstadt, Germany). For all *E. coli* heterologous expression constructs expression, cell lysis and GST purifica-tion were performed as described before[12]. For the comparison of inhibitory profiles of UmPit2 and UhPit2, after GST column purification, 4−5 flow throw

column-volumes per each protein were pooled and concentrated to a final volume of ca. 6 ml using the Vivaspin turbo 15 with 5 MWCO (Sartorius Stedim Biotech GmbH, Goettingen, Germany). Because further purification of UhPit2 using gel filtration was unsuccessful, a semi-quantitative analysis to calculate UmPit2 and UhPit2 protein concentration was performed using the basic protocol for SyproRuby protein gel staining (Invitrogen, Paisley, UK) and in-gel fluorescent scanning of 10 μl total protein loaded on gel. Signals were normalized by total protein concentration calculated using the Bradford method[45]. For stability experiments as well as for IC$_{50}$ values, UmPit2 was further purified using a gel filtration column (HiLoad Superdex 75 16/600, GE-Healthcare, Uppsala, Sweden) as previously described[12]. Protein concentration was measured at A280 nm in a 1 mm cuvette using a NanoDrop spectrophotometer (Thermo Scientific, Waltham, Massachusetts, USA). Prozip11 heterologous expression and purification was performed as previously described[15].

**Confocal microscopy.** Pit2/PID14 orthologs were transformed in *U. maydis* SG200 strains under the control of pit2 promoter and C-terminal mCherry tagged as described before[12]. *U. maydis* SG200 expressing internal mCherry under the control of *pit2* promoter was used as a negative control. Seven-day-old maize seedlings were inoculated with different strains expressing mCherry tagged pro-teins with an OD$_{600}$ = 2. Life cell imaging was performed at 2 dpi by Leica TCS SP8 using Laser confocal scanning microscope (Leica, Wetzlar, Germany). Image acquisition was conducted with a high-resolution CCD camera (C4742, Hama-matus). For life cell imaging of fungal hyphae in maize tissue, mCherry fluores-cence was excited with a 561 nm laser and emission was detected at 580–630 nm. Image data processing was accomplished using the Leica Application Suite software (Leica, Wetzlar, Germany).

**Protease activity assays using fluorogenic substrates.** Activity of PLCPs pre-sent in isolated apoplastic fluids was monitored using 10 μM final concentration of the fluorogenic substrate Z-Phe-Arg-7-amido-4-methylcoumarin (Sigma-Aldrich, Taufkirchen, Germany). AMC released (RFU) over time was monitored at 460 nm using a Tecan Infinite M200 PRO plate reader (Tecan Group Ltd., Männedorf, Switzerland). As a control apoplastic fluids were preincubated with 10 μM E-64 (Sigma-Aldrich, Taufkirchen, Germany) for 10 min. The inhibitor control was used to normalize all measurements and obtain a read-out for specific PLCP activity. Percentage of PLCP activity was calculated by setting up the sample without inhibitor to 100% activity. Isolation and ion-exchange chromatography fractiona-tion of apoplastic fluids was performed according to the method published by ref. [16]. UmPID14, StrepPID14, and UhPID14 were obtained as synthetic peptides with >98% purity from the company GenScript (New Jersey, USA). cMIP was obtained as synthetic peptide with >98% purity from the company Biomatik (Wilmington, Delaware, USA). Peptides were solubilized in water to a desired concentration. For papain activity measurements, 1 mg/ml papain from *Carica papaya* (Merck Millipore, Burlington, Massachusetts, USA) was activated at room temperature for 30 min in 60 mM sodium acetate pH 6 containing 10 mM DTT. 1.25 μg/ml final concentration was used for the experiments. 10 μM final con-centration of the fluorogenic substrate Z-Phe-Arg-AMC (Sigma-Aldrich, Tauf-kirchen, Germany) was used to monitor the activity of papain as described before for apoplastic fluids. Percentage of papain activity was calculated by setting up the sample without inhibitor to 100% activity. Percentage of papain inhibition was plotted against the inhibitor concentration. The activity of recombinant CP1 and CP2 proteins was monitored using 10 μM final concentration of the fluorogenic substrate Z-Leu-Arg-AMC (Sigma-Aldrich, Taufkirchen, Germany).

**Stability experiments.** Approximately 41 μg/ml heterologous expressed UmPit2 and UhPit2 proteins (concentration calculated with the previously described semi-quantitative approach based on SyproRuby) were incubated with or without apoplastic fluids (ca. 200 μg/ml) over time. Reaction was stopped by adding 2× SDS-gel loading buffer) and samples were analyzed by gel electrophoresis and SyproRuby staining (Invitrogen, Paisley, UK). Inhibition of apoplastic fluids con-taining specific inhibitors was performed as described in the main text. E-64 and pepstatin A were diluted in DMSO at a desired concentration. DCI was diluted in dimethyl formamide. All compounds have been purchased by Sigma-Aldrich (Taufkirchen, Germany). Stability experiments using F-24 were performed using ca. 1.4 μg/μl of the fraction and 3.2 μM of heterologous expressed UmPit2 or Prozip1.

**F-24 shotgun proteomics analysis.** F24 fractions were adjusted to 3 M Guani-diniumhydrochloride and carbamidomethylated using DTT/IAA. Samples were precipitated with 10% v/v trichloroacetic acid (TCA) o/N at 4 °C, followed by two subsequent washes with 100% ice-cold methanol. Air-dried pellets were resus-pended in 100 mM HEPES and digested with MS-grade Trypsin for 3 h at 42 °C. Sample cleanup was performed using C18 StageTips[46]. Separation was performed with a binary gradient from 5 to 32% B for 90 min with a total runtime of 2 h per sample on a nano-HPLC system (Ultimate 3000 nano-RSLC, Thermo) operated in a two-column setup (Acclaim PepMap 100 C18, ID 75 μm, trap column length 2 cm, particle size 3 μm, analytical column length 15 cm, particle size 2 μm, Thermo) coupled online to a high-resolution Q-TOF mass spectrometer (ImpactII, Bruker)

as described by ref. [47]. The Bruker HyStar Software (v3.2) was used to acquire line-mode MS spectra in a mass range from 200 to 1750 m/z at an acquisition rate of 4 Hz and the Top17 most intense ions were selected for fragmentation. The Max-Quant software package, v1.6.0.16[48] was used to identify peptides and proteins with the help of the UniProt *Zea mays* canonical protein dataset (UniProt Consortium, 2018), downloaded on April 2018 (39442 entries) and default settings for LFQ quantification. Validation of protein identification required at least two unique peptides for each protein and LFQ quantification in two out of three biological replicates. A list with the first most prominent 15 proteases identified in F-24 is summarized in Supplementary Data 1.

**In-gel digest of recombinant UhPit2**. UhPit2 was heterologous expressed in *E. coli* cells and protein samples were analyzed using gel electrophoresis. Coomassie-stained gel bands of putative recombinant UhPit2 were subjected to in-gel digestion with trypsin as described before[49]. Mass spectrometry analysis was performed as previously described using the UhPit2 sequence and canonical *Escherichia coli* database, downloaded on October 2015 (4305 entries).

**Pit2 and PID14 peptide MS analysis**. Recombinant UmPit2 (1 μg) and PID14 peptide (20 μM) were incubated with F24 fractions for 30 min at RT. Samples were preincubated with E-64 for inhibition of PLCPs or alternative inhibitor mix (EDTA 1 mM, Pepstatin A 30 μM, DCI 30 μM) for 10 min. After addition of final 3 M guanidinium hydrochloride, samples were precipitated with 10% v/v TCA o/N at 4 °C. Peptide fraction was recovered as supernatants by centrifugation at 21,500 × *g* for 15 min at 4 °C. Supernatants were purified by C18 StageTips[46] and injected onto a nano-HPLC system (Ultimate 3000 nano-RSLC, Thermo) operated in a two-column setup (Acclaim PepMap 100 C18, ID 75 μm, trap column length 2 cm, particle size 3 μm, analytical column length 50 cm, particle size 2 μm, Thermo) coupled online to a high-resolution Q-TOF mass spectrometer (ImpactII, Bruker). Peptides were separated with a binary gradient from 5 to 40% B for 40 min (A: $H_2O$ + 0.1% FA, B: ACN + 0.1% FA), followed by washing and re-equilibration steps to a total runtime of 1 h per sample. Line-mode MS spectra were acquired in a mass range from 150 to 2000 m/z at an acquisition rate of 4 Hz. For each MS spectrum, the Top14 most intense ions were selected for fragmentation. MaxQuant was used to identify peptides with the help of the UniProt *Zea mays* protein database, additionally supplemented with the PID14 or recombinant UmPit2 FASTA sequences. Validation of the peptide identifications required at least a value for quantification in two out of three biological replicates.

**Proteomic identification of protease cleavage sites**. Proteome-derived peptide libraries were generated by digesting *E. coli* K12 lysates with trypsin or GluC as described[50]. Two identical aliquots of each proteome-derived peptide library were preincubated for 10 min with a selected set of inhibitors (1 mM EDTA, 50 μM Pepstatin A, 50 μM DCI) before incubation with or without an aliquot of fraction F24 overnight at RT. The reaction was stopped by addition of guanidinium hydrochloride to a final concentration of 3 M. Peptides were duplex stable isotope labeled by reductive dimethylation. Isotopically light formaldehyde ($CH_2O$) and sodium cyanoborohydride ($NaBH_3CN$) was used to label control-treated library peptides and heavy formaldehyde ($^{13}CD_2O$) and sodium cyanoborohydride ($NaBH_3CN$) were used to label library aliquots after F24 incubation. Dimethylation was performed for 2 h at RT and quenched by addition of 100 mM Tris-HCl pH 7.5 for 1 h. Subsequently samples were mixed, desalted and purified by C18 StageTips[46].

One microgram desalted peptides was separated using a nano-HPLC system coupled online to a high-resolution Q-TOF mass spectrometer (ImpactII, Bruker). Peptides were eluted with a binary gradient from 5 to 35% B for 90 min (A: 0.1% formic acid, B: 0.1% formic acid in acetonitrile), followed by washing and re-equilibration steps to a total runtime of 2 h per sample. The Bruker HyStar Software (v3.2) was used to acquire line-mode MS spectra. in a mass range from 200 to 1750 m/z at an acquisition rate of 4 Hz. For each MS spectrum, the Top17 most intense ions were selected for fragmentation and an exclusion window of 40 s was applied.

Peptides were identified and quantified from the acquired MS spectra using the MaxQuant software package, v1.6.0.16[48] with PSM and protein FDRs set to 0.01 using the UniProt *E. coli* K12 proteome library (downloaded Nov 2015, 4313 entries). Trypsin was set as digestion enzyme for semi-specific searches (e.g. only one side of the peptide was required to match the trypsin specificity). Label multiplicity was set to two, considering light dimethylation (+28.0313 Da) and heavy dimethylation (+34.0631 Da) as peptide N-terminal and lysine labels. Carbamidomethylation of cysteine residues (+57.0215 Da) was set as fixed modification, methionine oxidation (+15.9949 Da) and protein N-terminal acetylation (+42.0106 Da) was considered as variable modifications.

Identified peptides that showed at least a fourfold increase in intensity after protease treatment compared to the control treatment or were exclusively present in the protease-treated condition were considered as putative cleavage products. An in-house developed Perl script was used to remove putative library peptides (trypsin specificity on both sides of the identified peptide) and to reconstruct the full cleavage windows from the identified cleavage products as described[29]. Aligned

validated cleavage windows were visualized as iceLogos[31], displaying site-specific differential amino acid abundance calculated as per cent difference compared to the *E. coli* K12 proteome as reference set (*p* value = 0.05).

**Activity-based protein profiling**. SA-treated apoplastic fluids were labeled with 0.2 μM MV202[28] in a reaction buffer containing 60 mM sodium acetate pH 6 and 10 mM DTT. Preincubation time was performed at room temperature and concentration of the inhibitors was used as described in the main text. Reaction was stopped by adding 2× SDS-gel loading buffer[51]. Samples were heated at 95 °C for 5 min and separated on a 12% SDS gel. Labeled proteins were visualized by in-gel fluorescence scanning using the ChemiDoc (Bio-Rad, California, USA) with the settings for Rhodamine (Ex/Em: 532/580 nm).

**Motif search by MEME and hidden Markov model building**. To determine conserved motifs in 11 Pit2-like sequences identified using blast analysis, we performed MEME[32] with the following parameters: mod = zoops, nmotifs = 30, minw = 6, and maxw = 40. From the MEME search results, the most significant motif (Motif No. 1) was used to make the HMM for further analysis. Sequences aligned to this motif were extracted in Stockholm format and used for hmmbuild program of HMMER v3.1b1.

**Statistical analysis**. ANOVA analysis was performed using the package Agricolae (v.1.2-4) in R Studio (v.1.1.383) performing a random model for the analysis of variance for disease rating of *U. maydis* inoculations and the activity of apoplastic PLCPs during infection. Disease index (DI) was calculated by multiplying the number of plants with certain symptom (category) by the given values to categories as follows: death plant = 11, heavy tumor on base of stem or stunted growth = 9, tumor on leave and/or steam = 7, small tumors = 5, chlorosis = 3 and without symptoms = 1. This number has been divided by the total number of plants used per strain. At least three biological replicates per strain have been performed and the strain SG200 was included in all assays to allow direct comparison with different mutant strains. Significance has been assessed with $\alpha = 0.05$.

Concentration range plots were performed in Origin using a nonlinear curve fit (dose response) and $\alpha = 0.05$. The $IC_{50}$ values of UmPit2 and UmPID14 were calculated for three biological replicates using the dose−response parameters. A one-way ANOVA for the $IC_{50}$ values with $\alpha = 0.05$ was performed.

**Reporting Summary**. Further information on experimental design is available in the Nature Research Reporting Summary linked to this article.

## Data availability

The mass spectrometry proteomics data have been deposited to the ProteomeXchange Consortium via the PRIDE[52] partner repository with the following dataset identifiers: PXD010582 for maize AF F24 shotgun proteome data, PXD010596 for AF F24 PICS profiling data, PXD010583 for UmPit2 processing data, and PXD010584 for PID14 processing data. Accession codes used in the manuscript are available under the following links: 5XRN, 1483958, SRP109982, NQLW00000000. The source data underlying Figs. 1a, 2a–d, 6d, h, 7c and Supplementary Figs. 1a, 5d are provided as a source data file. Other data supporting the findings of this study that are not directly available within the paper (and its supplementary information files) will be available from the corresponding author (G.D.) upon reasonable request.

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

## Acknowledgements

We would like to thank Gabriela Ronquillo-López for help in the statistical analysis, Christian Pichlo for proving the libraries for PICS and Jasper Depotter for his support in the annotation analysis of cMIP containing proteins. We thank Renier van der Hoorn and Jiorgos Kourelis for reading the manuscript and helpful comments and suggestions. Many thanks to Sophien Kamoun for fruitful discussions and scientific input. This work has been supported by the DFG via project DO1421/5-1 and the Cluster of Excellence on Plant Science (CEPLAS). P.F.H. acknowledges funding from the European Research Council under the European Union's Horizon 2020 research and innovation program (starting grant ProPlantStress, ID 639905).

## Author contributions

J.C.M.V. wrote the manuscript with input from all authors. G.D., J.C.M.V. and A.N.M. designed the experiments. J.C.M.V., A.N.M., U.M., B.Ö., M.B. and H.D. performed the functional characterization of Pit2. J.S.H. worked on the biochemical characterization of Pit2/PID. J.W. analyzed the sequence conservation of PID14/cMIP. F.D. and P.F.H. performed and analyzed PICS assays and mass spectrometry experiments.

## Additional information

**Competing interests:** The authors declare no competing interests.

