## [Peer Review File · Nature Communications]

Reviewers' comments:

Reviewer #1 (Remarks to the Author):

This manuscript is a nice continuation of previous findings by the Doehlemann group on plant PLCPs and their inhibition by the fungal effector Pit2. In this paper, the authors show convincing evidence that the full-length Pit2 effector is a substrate of maize PLCPs. They then suggested that processing of Pit2 by maize PLCPs released a more potent PLCP inhibitor (compared to full-length Pit2) from the PID14 domain. Furthermore, PID14 is a conserved motif found in other fungal pathogens and some bacteria that associate with plants. The authors then proposed that PID14 is a conserved microbial inhibitor of proteases (cMIP) that might be embedded in a PLCP substrate.

Overall, this is a well written paper with thorough experimentation and solid data. The biochemistry showing PLCP inhibition is beautiful. The finding that Pit2 is actually a substrate of the enzymes that it inhibits is intriguing and interesting. While I support the publication of this nice piece of research, some improvement is required.

Major points:

1) I don't feel the authors' claim that Pit2 "acts as (a) molecular mimicry" (in title and in the main text) is sufficiently supported by the data. What does Pit2 mimic? An endogenous plant substrate? The authors stated multiple times in the manuscript that Pit2 is "an excellent substrate". It is going to be hard to define "excellent" - maybe they can compare Pit2 with ProZIP1, a plant substrate that the same group reported in the recent Nature Plant paper for degradation efficiency by PLCP? Or maybe the authors could just tune this down.

2) An interesting question is why the fungal pathogen uses full-length Pit2 rather than just PID14. The idea that the full-length Pit2 is a better substrate to attract PLCPs, and then the PID14 is a better inhibitor to inhibit their activity is intriguing. But what the authors need to show is that Pit2 is indeed a better substrate than PID14. In Fig 2A, UmPID14 only partially complemented the deletion of Pit2, which could be supportive to this idea. However, they discussed that this might be due to insufficient secretion of the peptide (line 310-312). The authors also discussed the possibility of "structural conformational changes that are required for full inhibitory activity", which is unlikely since they showed that PID14 had a stronger inhibitory effect than full-length Pit2. So is the lower virulence activity of PID14 attributed to: 1) PID14 is not sequestering PLCPs as effectively as Pit2; OR 2) PID14 is not secreted by the pathogen as robustly as Pit2? I also hope the authors could include a positive control in this experiment (Fig 2A) to show that UmPit2 could completely complement the disease symptoms (so the partial complementation by UmPID14 was not an artifact).

3) Another interesting idea is that the fungal pathogens may package an inhibitory peptide in a host-specific surface structure. The authors explored this idea by comparing UmPit2 and UhPit2, which is very nice. I suggest the authors to take full advantage of the chimeras that they constructed to better support this notion. It would be nice to examine whether UmPit2-UhPID14 is still a substrate of maize PLCPs (i.e. can be degraded by AF). It would be even more interesting to examine whether the other chimera UhPit2-UmPID14 can be degraded similar to full-length UmPit2 - this may provide insight into why UhPit2-UmPID14 had a reduced virulence function compared to full-length UmPit2 (Fig 2D). Again, there was no full-length UmPit2 in this experiment, so it is hard to compare UhPit2-UmPID14 to UmPit2.

4) Clarify whether PID14 or part of PID14 is released or stay bound with PLCPs after the full-length Pit2 is processed. Both have been stated in different places of the manuscript (e.g. Line 342-343 and Line 359-360). I understand it is difficult to demonstrate either way and I don't think it is necessary to make a claim at this point. However, the authors should take caution when making speculations not supported by the current data.

5) In Fig 6F-6H, I could see UmPit2 and UmPID14 inhibited the activation of CP1, but I am confused on what happened with CP2, whose activity seemed to be enhanced. Please clarify and provide more convincing data if UmPit2/PID14 is an inhibitor of CP2.

6) I appreciate the authors' attempt to examine the PID14-like motif as a universal cMIP. Please

comment on what the cMIP-containing proteins in bacteria and other fungi are. For example, are they also secreted proteins? More importantly, the papain inhibition assay presented in Fig 7D for StrepPID14 is not convincing. There was hardly any inhibition by StrepPID14. Maybe a higher concentration(s) of StrepPID14 should be used? It would be nice to include a control peptide with the conserved R and W residues mutated StrepPID14 especially when the inhibitory effect of wild-type peptide is not that obvious. Also, statistical analysis should be included.

Some minor comments:

1. What is the * labeling in Figure 4B?
2. Fig 7F needs a control showing the expression of UmPit2(RW) mutant was similar to wild-type UmPit2 in the fungus. Same control is required for the other virulence assays, such as in Fig 1B, 2D.
3. In several places, the authors talked about "affinity" of Pit2 or PID14 to maize PLCPs. I wonder what does "affinity" mean – inhibitory effect or binding activity. Please clarify.
4. Line 146, "F-24 was pre-treatment with..." should be "pre-treated".
5. Line 269, "in a concentration manner..." should be "concentration-dependent".

Reviewer #2 (Remarks to the Author):

The fungus effector Pit2 of *Ustilago maydis* (Um) has inhibitory activity over papain-like cysteine proteases (PLCPs). The present manuscript by Misas-Villamil reveals an intriguing novel mechanism of action of this effector. The authors show that although the Pit2 effector is widely conserved, the full-length precursor Pit2 from different smut fungi does not fully restore pathogenicity of Pit2-deficient *U. maydis*. Furthermore, a chimeric UmPit2 construct carrying the PID14 domain from *U. hordei* (Uh) could not complement this mutant, while chimeric UhPit2 carrying UmPit2 restored pathogenicity as did complementation with UmPID14 alone. This genetic and phenotypic analysis is complemented by extensive biochemical work that demonstrates that both UmPit2 and UhPit2 are able to inhibit maize PLCPs, that the PID domain is in both cases a much more potent inhibitor than the full-length precursors, and that PLCPs themselves are able to process Pit2. Finally, the authors demonstrate that PID domain is not only conserved in smut fungi, but also several pathogenic and commensal bacteria and that the bacterial PID domains are functional PLCP inhibitors. The authors therefore suggest that PID domains could represent an ancient mechanism to suppress plant immunity by PLCP inhibition. The study addresses an important problem in plant sciences, is well designed and clearly described. The very sound quality extends to the proteomic data and proteomic assays. The interesting new plant effector mechanism appears to be widely conserved and should therefore find wide interest in the broad readership of *Nature Communications*. However, a few points could be addressed to further improve the study:

1. Evidence that cleavage of Pit2 by PLCPs is necessary for the inhibitory activity is rather indirect. Does impaired Pit2 cleavage prevent activation of the inhibitory activity? This could e.g. be tested by point mutation of the suggested PLCP cleavage sites in Pit2.
2. It is suggested that the inefficient complementation of Pit2-deficient *U. maydis* by UhPit2 could reflect a host-specific adaptation or that the effectors may have acquired a different function (P4, line 87/88; p10 line 288/9). It would be good if the authors could demonstrate that the basic mechanism – proteolytic release of the inhibitory domain by the target enzyme – is conserved in these effectors, e.g. using UhPID14 and barley PLCPs.
3. Fig 6: Preincubation with UmPit2 and UmPID14 suppresses CP1a/b labeling, but does not affect CP2 labeling. This suggests that the CP1 enzymes are the preferential targets of UmPit2/UmPID14. Have the authors further explored this differential affinity, e.g. with recombinant enzymes or pull-down assays?
4. What is the preferential target of UhPID14 in this assay?

Minor comments:

1. P8, line 221ff, Fig 6: This is confusing. Is the apoplast fluid pre-incubated, then labeled, or is

the labeling time variable?

2.P10 line 277ff: "complemented Δ Pit2_Pit2 and Δ Pit2_Pit2R48AW49A showed no difference in tumor formation". This is confusing, as the mutated effector did not complement (Fig. Please rephrase.

3.Fig.7 is somewhat complicated and particularly the right hand side does not add much information as all proteins are eventually degraded by a variety of proteases.

Reviewer #3 (Remarks to the Author):

The manuscript by Misas-Villamil et al advances on previous work by this laboratory on the role of the Pit2 effector in *U. maydis*. In this study the authors demonstrate that the embedded peptide PID14 is more active as a PLCP inhibitor than is Pit2 whilst showing that the peptide is released by maize PLCPs. Inhibition studies suggested an important role for 2 amino acids in the activity of the released peptide. Finally, the authors show that Pit2-like proteins are present in related fungi as well some bacteria and appear to harbour a conserved function.

For the most part, the paper is well written and the conclusions are relatively well supported by the data. I am not entirely convinced that the findings presented are a sufficient and broad enough advance for this journal but that is not my decision to make. I do though like the concept of the molecular mimicry but there are a couple of issues that need to be addressed before this manuscript should be considered.

The obvious one to me is the difference in activity observed between the Pit2 and PID14 homologues from *Um* and other related fungi (specifically *Uh*). The authors allude to this in the discussion briefly but surely that likely reason for the difference is the fact that the pathogens infect different hosts. As such, I struggled with comments like "...we found its function being not conserved in the barley smut *Uh*". Unless the authors actually checked its function in barley, I am not sure that this can be claimed. Indeed, these would be obvious experiments to me to demonstrate a true advance to the field and compare the activities of the proteins and peptides on SA-induced barley apoplast. Given the protein is easily expressed, I imagine this would be a simple experiment.

A few other points:

1. P2L2. Wording/grammar issues "... food security and global food supply have become major challenges to address this century"

2. P2L4. Consider changing "... being fungal pathogens accountable of..." to "... with fungal pathogens being accountable for..."

3. P2L11. Instead of "... responses and so achieving a successful...", consider changing to "... responses and achieving successful ...".

4. P4L67. Is it correct to write the dominant form of the gene with a lower case first letter?

4. P4L66, 1st results section. The results certainly appear compelling that the different Pit2 genes are (largely) unable to complement the *Um*Pit2 mutation. However one does need to consider that these are technically heterologous expression experiments. Has the care been taken to ensure that these proteins are correctly translated/folded/stable in the heterologous host? If not, how can one be sure that the lack of complementation is due to functionality than rather protein expression issues?

5. Figure 5 legend typo, "input" rather than "imput"

6. P6L146. "pre-treated" not "pre-treatment"

7. Figure 6 legend, "reproducibly" not "reproducible"

8. P6L159. This isn't my area of expertise but I did struggle with the justification for the "putative docking site". I realise some subsequent modelling etc provided some circumstantial evidence but I think the authors could better justify at this point why precisely they consider that potential docking site (particularly for non-experts).

P6L164. A new paragraph from the sentence starting with "To ...".

P7L193. Could the authors briefly comment on the C-score of -3.35? Does this provide confidence in the model? Structural models do sometimes raise more questions than they answer.

P8L211. Could the authors comment on the differences in inhibition activity observed in Figures 3A/C and that presented in Figure 6E?

P10L276. In planta needs to be italicised.

P10L277. How do the authors know that the significant mutations made to the Arg and Trp (to Alanines) has not affected the protein folding/stability? These are not insignificant changes. There are many instances where minor changes to protein sequences have affected stability/folding resulting in an insoluble/degraded protein. In my opinion, the authors need to provide some evidence that the mutant protein being expressed in Um is actually present prior to making conclusions as to the importance of these 2 amino acids.

P12L337. Sentence starting with "As soon ..." is poorly worded and difficult to understand. Please consider revising.

Response to reviewer comments:

Reviewer #1:

This manuscript is a nice continuation of previous findings by the Doehlemann group on plant PLCPs and their inhibition by the fungal effector Pit2. In this paper, the authors show convincing evidence that the full-length Pit2 effector is a substrate of maize PLCPs. They then suggested that processing of Pit2 by maize PLCPs released a more potent PLCP inhibitor (compared to full-length Pit2) from the PID14 domain. Furthermore, PID14 is a conserved motif found in other fungal pathogens and some bacteria that associate with plants. The authors then proposed that PID14 is a conserved microbial inhibitor of proteases (cMIP) that might be embedded in a PLCP substrate.

Overall, this is a well written paper with thorough experimentation and solid data. The biochemistry showing PLCP inhibition is beautiful. The finding that Pit2 is actually a substrate of the enzymes that it inhibits is intriguing and interesting. While I support the publication of this nice piece of research, some improvement is required.

Major points:

1) I don't feel the authors' claim that Pit2 "acts as (a) molecular mimicry" (in title and in the main text) is sufficiently supported by the data. What does Pit2 mimic? An endogenous plant substrate? The authors stated multiple times in the manuscript that Pit2 is "an excellent substrate". It is going to be hard to define "excellent" - maybe they can compare Pit2 with ProZIP1, a plant substrate that the same group reported in the recent Nature Plant paper for degradation efficiency by PLCP? Or maybe the authors could just tune this down.

We thank the reviewer for this comment because it shows that we were not clear in some descriptions. In our opinion the term "molecular mimicry" is suitable to describe the mechanism of action of Pit2. Ronald & Joe described the molecular mimicry as a molecule produced by pathogens that resemble host factors such as substrates of host enzymes to suppress host immune responses, facilitate infection and maintain the biotrophic interaction (Ronald & Joe, 2018). We think that this definition fits quite well to the finding that Pit2 acts as a substrate for host PLCPs to suppress their activity. We now included a more detailed explanation to better clarify the term and why we are using it (lines 87ff).

Regarding the term "excellent substrate" we can fully agree with the reviewer. Indeed this is not a very useful term, and particularly not a precise quantitative description of the biochemical properties of the Pit2 effector. We therefore have changed this description to the term "suitable substrate".

Moreover, we have included new data supporting our claim of Pit2 acting as a molecular mimicry for maize PLCPs. To this end, we included the Prozip1 propeptide that was mentioned by this reviewer. Zip1 was recently published by our lab to be an apoplastic peptide that activates SA-associated defense responses in maize (Ziemann et al., 2018, Nature Plants). Zip1 released from the endogenous propeptide Prozip1 is mediated by the PLCPs CP1 and CP2, which are the same proteases that are targeted by Pit2. In a new experiment, we co-incubated Pit2 and Prozip1 with PLCP fraction-24 and found that, in a competition assay, both proteins are cleaved at similar rates. This suggests that Pit2 resembles is targeted by PLCPs as efficient as the endogenous signaling molecule Prozip1. Based on this result, one could even hypothesize that Pit2, which is highly expressed and accumulates in the apoplast, can outcompete the very low expressed Prozip1 in the native situation to prevent Zip1-release

and consequently avoid induction of SA-associated defense responses. This, however, remains speculative as we do not yet have the tools to verify this hypothesis *in-vivo* and therefore we only mention it in this letter.

The Prozip1 data is shown in the new additional figure 5F.

2) An interesting question is why the fungal pathogen uses full-length Pit2 rather than just PID14. The idea that the full-length Pit2 is a better substrate to attract PLCPs, and then the PID14 is a better inhibitor to inhibit their activity is intriguing. But what the authors need to show is that Pit2 is indeed a better substrate than PID14. In Fig 2A, UmPID14 only partially complemented the deletion of Pit2, which could be supportive to this idea. However, they discussed that this might be due to insufficient secretion of the peptide (line 310-312). The authors also discussed the possibility of “structural conformational changes that are required for full inhibitory activity”, which is unlikely since they showed that PID14 had a stronger inhibitory effect than full-length Pit2. So is the lower virulence activity of PID14 attributed to: 1) PID14 is not sequestering PLCPs as effectively as Pit2; OR 2) PID14 is not secreted by the pathogen as robustly as Pit2?

We are very thankful for this comment. Showing that Pit2 is a better substrate than PID14 is definitely a relevant point. To address this, we performed a competition experiment between the PLCP fluorogenic substrate Z-FR-AMC and Pit2 or PID14, respectively. When the fluorogenic substrate is cleaved it emits light, which is the read out for activity. Being substrates for the PLCPs therefore means that addition of either Pit2 or PID14 will result in reduced cleavage of the fluorogenic substrate, i.e. light emission will be lower proportional to the substrate preference. To unlink inhibitory activity and function as a substrate, we used mutated versions of both UmPit2 and UmPID14, which are lacking inhibitory activity. This experiment shows that both, Pit2 and PID14 are PLCP substrates, but Pit2 is cleaved at a significantly higher efficiency. This data has been added in the new supplementary figure S6.

To address the second part of this point, we have tested secretion of PID14 via microscopy, using mCherry tagged UmPID14 in comparison to UmPit2-mCherry. This experiment showed no differences in expression and localization of UmPit2 compared to UmPID14. We therefore consider it unlikely that the lower virulence activity of PID14 is due to affected expression or secretion. Main reasons for this phenotype might be mainly attributed to the poor molecular mimicry effect produced by PID14 in comparison to Pit2. The data is shown in new supplementary figure S1 and it is discussed in the text, lines 359ff

I also hope the authors could include a positive control in this experiment (Fig 2A) to show that UmPit2 could completely complement the disease symptoms (so the partial complementation by UmPID14 was not an artifact).

We have included the positive control (UmPit2 complementation strain) in Fig. 2A that fully complements symptoms similar to the SG200 control strain, confirming that UmPID14 partial complementation is not an artefact.

3) Another interesting idea is that the fungal pathogens may package an inhibitory peptide in a host-specific surface structure. The authors explored this idea by comparing UmPit2 and UhPit2, which is very nice. I suggest the authors to take full advantage of the chimeras that they constructed to better support this notion. It would be nice to examine whether UmPit2-

UhPID14 is still a substrate of maize PLCPs (i.e. can be degraded by AF). It would be even more interesting to examine whether the other chimera UhPit2-UmPID14 can be degraded similar to full-length UmPit2 – this may provide insight into why UhPit2-UmPID14 had a reduced virulence function compared to full-length UmPit2 (Fig 2D). Again, there was no full-length UmPit2 in this experiment, so it is hard to compare UhPit2-UmPID14 to UmPit2.

We performed the suggested stability experiments of the heterologous expressed chimeras in apoplastic fluids. UmPit2-UhPID14 is processed over time, while UhPit2-UmPID14 is stable. This suggests that the UmPit2 backbone, but not the UhPit2 backbone is recognized as a substrate, independent of the embedded PID14 sequence. It also suggests that both, cleavage and inhibition are necessary for full virulence function supporting the idea of UmPit2 molecular mimicry capacity. This data is shown in the new supplementary figure S7. UmPit2 complementation has been added to figure 2A and it is comparable to SG200 infection.

4) Clarify whether PID14 or part of PID14 is released or stay bound with PLCPs after the full-length Pit2 is processed. Both have been stated in different places of the manuscript (e.g. Line 342-343 and Line 359-360). I understand it is difficult to demonstrate either way and I don't think it is necessary to make a claim at this point. However, the authors should take caution when making speculations not supported by the current data.

We thank the reviewer for the understanding that it is difficult to demonstrate this by additional direct evidence. We have rephrased lines 391ff to be more cautious with our statement, which now reads: “We could not identify the full sequence of UmPID14 using MS analysis of UmPit2 incubated with F-24 (Fig. 6A). This could indicate that UmPID14 is further processed and it contains the inhibitory portion that stays bound to the PLCPs and therefore not free for MS identification”. We also rephrased lines 407ff as follows: “the inhibitory portion within the UmPID14 motif remains bound to the PLCP, resulting in inhibition”

5) In Fig 6F-6H, I could see UmPit2 and UmPID14 inhibited the activation of CP1, but I am confused on what happened with CP2, whose activity seemed to be enhanced. Please clarify and provide more convincing data if UmPit2/PID14 is an inhibitor of CP2.

We apologize for our unsatisfactory explanation of the experimental data shown in figure 6. We now have added a more detailed description to the labelling pattern observed in Fig 6 F – H and a new paragraph contained in lines 243ff.

Figure 6F shows a time course labelling of PLCPs using the probe MV202 without inhibitor. Samples were collected from one single reaction at different time points. One can see that the probe preferentially labels CP2, since at time 0 the strongest band to see is mainly CP2. After 120 min incubation of the AF with MV202 the labelling changes and the most intense band corresponds to CP1, whereas CP2 labelling decreases over time. MV202 binds covalent and irreversible to the active site of the PLCPs, so when a signal disappears this likely reflects degradation of the labelled protein. Therefore, this experiment suggests that CP1 is present in the AF but in an inactive state. Activation of CP1 occurs over time and this activation leads to degradation of CP2 (direct or indirect). Figure 6 G and H shows pre-incubation time with UmPit2 and UmPID14 but only one hour labelling with MV202, comparable to Fig. 6F 60 min incubation. Important point is: the whole picture changes when UmPit2 or UmPID14 are added to the sample. Both, UmPit2 and UmPID14 stabilize CP2 in the AF and both

molecules block the activation of CP1. Moreover, UmPID14 pre-incubation not only suppressed activation of CP1 but at longer time points it might also inhibit CP2.

In addition, this reviewer point motivated us to perform a new experiment to specifically test inhibition of CP1 and CP2 by UmPit2 and UmPID14. To this end, we used *N. benthamiana* expressed CP1 and CP2. The new experiment shows inhibition of both CP1 and CP2 by UmPit2. At lower concentrations, inhibition of CP2 seems to be more efficient, however, the differences are not significant and the data did not allow calculation of IC50 values.

Similarly, UmPID14 inhibits both CP1 and CP2 but here the data shows a significantly lower IC50 value for CP1 indicating a preference for its inhibition.

In summary, this experiment shows that CP1 and CP2 are both inhibited by UmPit2/UmPID14 and it indicates that PID14 is a more efficient inhibitor of CP1 than of CP2. The data is shown in new supplementary figure S8.

6) I appreciate the authors' attempt to examine the PID14-like motif as a universal cMIP. Please comment on what the cMIP-containing proteins in bacteria and other fungi are. For example, are they also secreted proteins?

We investigated the features of cMIP-containing proteins as suggested and we have added more details in the discussion section about the cMIP-containing proteins found in our analysis (lines 428ff).

Among the sequences that contain cMIP motif, there were four from Gram positive bacteria, as well as the Gram negative bacterium *Burkholderia vietnamiensis*. The sequence from the Gram positive Microbacterium was annotated as tripartite tricarboxylate transporter TctB family protein, and that from *B. vietnamiensis* was annotated as Xaa-Pro peptidase with cMIP motif forming a part of Creatinase N-terminal domain. Other bacterial sequences were annotated as hypothetical proteins. All fungal sequences with cMIP motif were also annotated as either hypothetical or unknown proteins. This information is summarized in supplementary Table S2.

More importantly, the papain inhibition assay presented in Fig 7D for StrepPID14 is not convincing. There was hardly any inhibition by StrepPID14. Maybe a higher concentration(s) of StrepPID14 should be used? It would be nice to include a control peptide with the conserved R and W residues mutated StrepPID14 especially when the inhibitory effect of wild-type peptide is not that obvious. Also, statistical analysis should be included.

We accept this criticism that the data shown in previous Fig 7D was not fully convincing. We therefore performed a new analysis and show a different presentation of the data. In the new Figure 7D we plotted the inhibition of papain against peptide concentration. One can see inhibition of papain by all three peptides tested. While both UmPID14 and StrepPID14 reach a maximal inhibition of about 70%, up to 90% of papain activity is inhibited by the cMIP peptide.

Due to the amount of peptides consumed by these experiments we were unable to reach higher concentrations which will allows us to come up with an IC50 value. Comparing the inhibitory efficiencies between peptides against papain using the IC50 values would have allowed us to perform the required statistical analysis. At this stage we can only conclude that the tested peptides inhibit papain.

Regarding “RW” mutant peptides we have to admit that due to technical problems it is not possible to include them in this experiment. Synthetic peptides carrying these mutations are not water soluble and therefore cannot be used in this papain inhibition assay.

Some minor comments:

1. What is the * labeling in Figure 4B?

E. coli contaminant from the protein purification. Information has been added to figure legend.

2. Fig 7F needs a control showing the expression of UmPit2(RW) mutant was similar to wild-type UmPit2 in the fungus. Same control is required for the other virulence assays, such as in Fig 1B, 2D.

We have added microscopy data to confirm production of all proteins used for the complementation in different virulence assays. Tested proteins were tagged with mCherry and protein production was tested in SG200 background to avoid altered expression in the mutants due to different levels of plant defense responses. mCherry fluorescence was compared to the mCherry control, which localizes mainly in the cytoplasm. Line blots of mCherry signals show similar secretion of all tested Pit2/PID14 proteins from biotrophic *U. maydis* hyphae. These data are shown in the new supplementary figure S1.

3. In several places, the authors talked about “affinity” of Pit2 or PID14 to maize PLCPs. I wonder what does “affinity” mean – inhibitory effect or binding activity. Please clarify.

We thank the reviewer for pointing this out. We prefer to use the term “affinity” to describe the inhibitory effect but indeed it can be misleading. As a substrate Pit2 has higher “binding capacity” but as an inhibitor PID14 has higher affinity. To avoid misunderstandings, those terms have been carefully revised and change in the manuscript.

4. Line 146, “F-24 was pre-treatment with...” should be “pre-treated”.

This has now been corrected.

5. Line 269, “in a concentration manner...” should be “concentration-dependent”.

This has now been corrected.

Reviewer #2:

The fungus effector Pit2 of *Ustilago maydis* (Um) has inhibitory activity over papain-like cysteine proteases (PLCPs). The present manuscript by Misas-Villamil reveals an intriguing novel mechanism of action of this effector. The authors show that although the Pit2 effector

is widely conserved, the full-length precursor Pit2 from different smut fungi does not fully restore pathogenicity of Pit2-deficient *U. maydis*. Furthermore, a chimeric UmPit2 construct carrying the PID14 domain from *U. hordei* (Uh) could not complement this mutant, while chimeric UhPit2 carrying UmPit2 restored pathogenicity as did complementation with UmPID14 alone. This genetic and phenotypic analysis is complemented by extensive biochemical work that demonstrates that both UmPit2 and UhPit2 are able to inhibit maize PLCPs, that the PID domain is in both cases a much more potent inhibitor than the full-length precursors, and that PLCPs themselves are able to process Pit2. Finally, the authors demonstrate that PID domain is not only conserved in smut fungi, but also several pathogenic and commensal bacteria and that the bacterial PID domains are functional PLCP inhibitors. The authors therefore suggest that PID domains could represent an ancient mechanism to suppress plant immunity by PLCP inhibition. The study addresses an important problem in plant sciences, is well designed and clearly described. The very sound quality extends to the proteomic data and proteomic assays. The interesting new plant effector mechanism appears to be widely conserved and should therefore find wide interest in the broad readership of Nature Communications. However, a few points could be addressed to further improve the study:

1. Evidence that cleavage of Pit2 by PLCPs is necessary for the inhibitory activity is rather indirect. Does impaired Pit2 cleavage prevent activation of the inhibitory activity? This could e.g. be tested by point mutation of the suggested PLCP cleavage sites in Pit2.

We thank the reviewer for this interesting question. We took advantage of the chimeras we have generated to better understand if cleavage of Pit2 is necessary for the inhibitory activity. Stability experiments of the chimeras showed that the chimera UhPit2_UmPID14 is stable (new supplementary Fig. S7 A). At the same time it leads to partial, but not complete restoration of *Ustilago* tumor formation (Fig. 2D). On the other hand, the chimera UmPit2_UhPID14 can be processed (new supplementary Fig. S7 B), but it cannot complement virulence (Fig. 2D). This implies that cleavage alone is not sufficient for the inhibitory activity, but it is required for full virulence.

To address if impaired cleavage correlates with inhibition of PLCPs we have measured the PLCP activity of maize leaves after infection with *U. maydis* strains expressing the chimera mutants vs native Pit2. This revealed a slightly increased PLCP activity in samples with UhPit2-UmPID14, compared to native Pit2. Chimera UmPit2-UhPID14 results in higher PLCP activity that is only exceeded in samples infected with the Δ pit2 mutant (new supplementary Fig. S7 C). These PLCP activities correlate with the virulence phenotype of the respective mutants, shown in Fig. 2D. Altogether, these data indicate that cleavage is necessary for virulence, but it is not required for full inhibition.

2. It is suggested that the inefficient complementation of Pit2-deficient *U. maydis* by UhPit2 could reflect a host-specific adaptation or that the effectors may have acquired a different function (P4, line 87/88; p10 line 288/9). It would be good if the authors could demonstrate that the basic mechanism – proteolytic release of the inhibitory domain by the target enzyme – is conserved in these effectors, e.g. using UhPID14 and barley PLCPs.

We also followed this suggestion by the reviewer and have performed treatment of barley leaves with / without salicylic acid and isolated apoplastic fluids of treated leaves. We also produced UhPit2 in *E. coli* and used recombinant protein for stability experiments with isolated apoplastic fluids (new supplementary figure S10). This incubation of UhPit2 with

barley treated AF fluids did not show cleavage or degradation of UhPit2 over time, suggesting that the substrate mimicry mechanism seen for UmPit2 in maize is actually not conserved in barley.

This is also in line with the observation that in barley PLCPs are not activated by SA (new figure S9), which suggests a different signalling mechanism in barley compared to maize. In turn, this suggests that Pit2 effectors of *U. maydis* and *U. hordei* might have undergone host-specific adaption leading to different functional mechanisms.

3. Fig 6: Preincubation with UmPit2 and UmPID14 suppresses CP1a/b labeling, but does not affect CP2 labeling. This suggests that the CP1 enzymes are the preferential targets of UmPit2/UmPID14. Have the authors further explored this differential affinity, e.g. with recombinant enzymes or pull-down assays?

We thank the reviewer for this comment. At first, we have added a better explanation of the activity blots showed in Fig.6 (lines 243ff), where indeed suppression of CP1 might stabilize CP2 thus inhibiting a cascade of degradation. Moreover we have produced recombinant CP1 and CP2 in *N. benthamiana* and tested the inhibitory activity of both UmPit2 and UmPID14 on the two proteases.

As elaborated in response to reviewer 1 (point 5), we found that both, CP1 and CP2 are inhibited by UmPit2 (new supplementary Figure S8 A-B). At lower concentrations, inhibition of CP2 seems to be more efficient, however, the differences are not significant and the data did not allow calculation of IC50 values. Similarly, UmPID14 inhibits both CP1 and CP2 but here the data shows a significantly lower IC50 value for CP1 indicating a preference for its inhibition.

In summary, this experiment shows that CP1 and CP2 are both inhibited by UmPit2/UmPID14 and it suggests that CP1 is the preferential target of UmPID14.

4. What is the preferential target of UhPID14 in this assay?

To address this question, we included UhPID14 in the inhibition assays discussed in the point above (supplementary figure S2). However, unlike for UmPID14 there seems to be no preferential target of UhPID14 in this assay, although CP1 seems to be more efficiently inhibited at high concentrations.

Minor comments:

1.P8, line 221ff, Fig 6: This is confusing. Is the apoplastic fluid pre-incubated, then labeled, or is the labeling time variable?

We apologize for our imprecise explanation of this figure. In figure 6F the apoplastic fluid is labelled at different time points, this shows us the dynamics of PLCP activation/degradation over time. Labeling in Fig. 6F has been adjusted. In figure 6G and 6H the apoplastic fluid was first pre-incubated at different time points and then labelled for 1 hour. Those experiments allow the inhibitors UmPit2 or UmPID14 to achieve suppression of PLCPs before the probe is added. To avoid misunderstandings, we now added a more detailed description of the experiments (lines 243ff) and labelled the gels accordingly.

2.P10 line 277ff:”complemented \square Pit2_Pit2 and \square Pit2_Pit2R48AW49A showed no

difference in tumor formation”. This is confusing, as the mutated effector did not complement (Fig. Please rephrase.

We have rephrased the sentence (lines 318ff) by “comparison of disease rating at 12 dpi between SG200 and the complemented Δ Pit2_Pit2 strain did not show differences in virulence (Fig. 7F). In contrast, the Δ Pit2_Pit2^{R48AW49A} strain cannot complement tumor formation resembled the avirulent phenotype of the Δ Pit2 strain (Fig. 7F). This experiment confirms that the conserved RW residues are essential for Pit2 virulent function *in planta*.”

3.Fig.7 is somewhat complicated and particularly the right hand side does not add much information as all proteins are eventually degraded by a variety of proteases.

We have modified the model according to the reviewers comment and we hope that it now provides a better visualization of the proposed mechanism.

Reviewer #3:

The manuscript by Misas-Villamil et al advances on previous work by this laboratory on the role of the Pit2 effector in *U. maydis*. In this study the authors demonstrate that the embedded peptide PID14 is more active as a PLCP inhibitor than is Pit2 whilst showing that the peptide is released by maize PLCPs. Inhibition studies suggested an important role for 2 amino acids in the activity of the released peptide. Finally, the authors show that Pit2-like proteins are present in related fungi as well some bacteria and appear to harbour a conserved function.

For the most part, the paper is well written and the conclusions are relatively well supported by the data. I am not entirely convinced that the findings presented are a sufficient and broad enough advance for this journal but that is not my decision to make. I do though like the concept of the molecular mimicry but there are a couple of issues that need to be addressed before this manuscript should be considered.

The obvious one to me is the difference in activity observed between the Pit2 and PID14 homologues from *Um* and other related fungi (specifically *Uh*). The authors allude to this in the discussion briefly but surely that likely reason for the difference is the fact that the pathogens infect different hosts. As such, I struggled with comments like “...we found its function being not conserved in the barley smut *Uh*”. Unless the authors actually checked its function in barley, I am not sure that this can be claimed. Indeed, these would be obvious experiments to me to demonstrate a true advance to the field and compare the activities of the proteins and peptides on SA-induced barley apoplast. Given the protein is easily expressed, I imagine this would be a simple experiment.

We thank the reviewer for this comment and at the same time apologize for being imprecise in our wording. We have modified the respective parts in the introduction and discussion and added recent knowledge about *U. maydis* effectors and how its unique life style can result in functional diversification of effectors. A recent example is the Tin2 effector. It is produced by the two closely related smuts *U. maydis* and *S. reilianum*. Although both fungi infect maize, Tin2 obtained differential functions in the two pathogens (Tanaka et al., 2018).

Moreover, we followed the reviewer's advice and performed additional experiments on *U. hordei* Pit2 and barley PLCPs (see also Reviewer 2, point 2). In a nutshell, barley apoplastic PLCPs are not activated by SA as it is seen in maize (Fig. S9). Similarly, UhPit2 is neither cleaved by barley PLCPs, nor by maize PLCPs (Fig. S10 and Fig. 4B). All these data indicate that barley and maize evolved different apoplastic signalling cascades. Consequently, we consider it likely that in *U. hordei* the Pit2 effector might have adapted to its host and evolved p a different function.

Apart from this, we need to admit that a comprehensive characterization of apoplastic SA-signaling in barley and its modulation by *Ustilago hordei* (by a yet unknown mechanism) is an own project beyond the scope of this manuscript.

A few other points:

1. P2L2. Wording/grammar issues "... food security and global food supply have become major challenges to address this century"

It has been now rephrased.

2. P2L4. Consider changing "... being fungal pathogens accountable of..." to "... with fungal pathogens being accountable for..."

It has been now corrected.

3. P2L11. Instead of "... responses and so achieving a successful...", consider changing to "... responses and achieving successful ...".

It has been now corrected.

4. P4L67. Is it correct to write the dominant form of the gene with a lower case first letter?

Is it widely accepted in our community to use lower case italic letters for fungal genes (*pit2*), while proteins are not italic and only the first letter in upper case (Pit2). Gene deletions are indicated by "Δ" ($\Delta pit2$). We are following this nomenclature consistently in all our manuscripts.

4. P4L66, 1st results section. The results certainly appear compelling that the different Pit2 genes are (largely) unable to complement the UmPit2 mutation. However one does need to consider that these are technically heterologous expression experiments. Has the care been taken to ensure that these proteins are correctly translated/folded/stable in the heterologous host? If not, how can one be sure that the lack of complementation is due to functionality than rather protein expression issues?

We appreciate the reviewer pointing this out, as it is indeed relevant to confirm that Pit2 versions that cannot complement *U. maydis* virulence are actually produced (and secreted). We therefore tested all Pit2/PID14 versions that were used for expression in *U. maydis*. To this end, we expressed mCherry tagged versions of all proteins in *U. maydis* (see also Reviewer 1, point 2). All strains were infected to maize seedlings and analyzed by confocal microscopy. This experiment showed indistinguishable mCherry signals for native UmPit2

and all tested Pit2/PID14 versions, confirming proper production and secretion of the proteins by *U. maydis* *in vivo*. This data is shown in the new supplementary figure S1.

5. Figure 5 legend typo, “input” rather than “imput”

It has been now corrected.

6. P6L146. “pre-treated” not “pre-treatment”

It has been now corrected.

7. Figure 6 legend, “reproducibly” not “reproducible”

It has been now corrected.

8. P6L159. This isn't my area of expertise but I did struggle with the justification for the “putative docking site”. I realise some subsequent modelling etc provided some circumstantial evidence but I think the authors could better justify at this point why precisely they consider that potential docking site (particularly for non-experts).

A more careful explanation about our proposal of a docking site has been added (lines 172ff). We hope the reviewers and the readers can understand that finding direct evidence for binding of the specific inhibitory portion to the PLCPs is highly challenging, which is a reason why we propose to name of that binding surface as the docking site.

P6L164. A new paragraph from the sentence starting with “To ...”.

It has been now corrected.

P7L193. Could the authors briefly comment on the C-score of -3.35? Does this provide confidence in the model? Structural models do sometimes raise more questions than they answer.

C-score is a confidence score for estimating the quality of predicted models by I-TASSER. It is calculated based on the significance of threading template alignments and the convergence parameters of the structure assembly simulations. C-score is typically in the range of [-5 to 2], where a C-score of higher value indicates a model with a high confidence and vice-versa. The score of -3.35 was the best score calculated for Pit2 out of five models. This structure has been also predicted using other programs such as Phyre2 with similar results.

P8L211. Could the authors comment on the differences in inhibition activity observed in Figures 3A/C and that presented in Figure 6E?

We thank this reviewer to point out this mistake. The scales in figure 3 are different as in figure 6E. We have corrected this mistake and made all figures with Log inhibitor concentrations. On the other side, inhibition curves in figure 3A are made with proteins lacking one purification step (gel filtration). Reason for it is technical difficulties with the

second purification step of UhPit2. Therefore both, UmPit2 and UhPit2 protein concentrations has been adjusted using SyproRuby fluorescent quantification. This method has been explained in the material and methods part. Curves in figure 6E are performed with gel-filtration purified UmPit2 protein to allow comparison with UmPID14 and calculation of IC50 values.

P10L276. *In planta* needs to be italicised.

It has been now corrected.

P10L277. How do the authors know that the significant mutations made to the Arg and Trp (to Alanines) has not affected the protein folding/stability? These are not insignificant changes. There are many instances where minor changes to protein sequences have affected stability/folding resulting in an insoluble/degraded protein. In my opinion, the authors need to provide some evidence that the mutant protein being expressed in Um is actually present prior to making conclusions as to the importance of these 2 amino acids.

As stated above (point 4), this has been tested using confocal microscopy. The Pit2^{R48AW49A} mutant has been included in this analysis, confirming expression and secretion of the protein *in-planta* (new supplementary figure S1).

P12L337. Sentence starting with “As soon ...” is poorly worded and difficult to understand. Please consider revising.

Sentence has been re-phrased.

REVIEWERS' COMMENTS:

Reviewer #1 (Remarks to the Author):

The authors have sufficiently addressed all of my previous concerns. I continue to believe this is a very nice paper and support its publication.

Wenbo Ma

Reviewer #2 (Remarks to the Author):

All my comments have been adequately addressed. I congratulate the authors on their interesting work and strongly support publication in Nat Comm

Reviewer #3 (Remarks to the Author):

This is a revised submission of the previous version submitted by Misas-Villamil et al further examining the role of the Pit2 effector in *U. maydis*. I was generally quite satisfied with the original submission and I felt that it did significantly contribute to the field. I did though also feel that a few issues needed to be clarified (which I note some of which were also picked up by the other reviewers). I am happy to see that the authors have taken these concerns seriously and in my opinion, have adequately addressed each of these.

Reviewer comments response letter

REVIEWERS' COMMENTS:

Reviewer #1 (Remarks to the Author):

The authors have sufficiently addressed all of my previous concerns. I continue to believe this is a very nice paper and support its publication.

Wenbo Ma

Reviewer #2 (Remarks to the Author):

All my comments have been adequately addressed. I congratulate the authors on their interesting work and strongly support publication in Nat Comm

Reviewer #3 (Remarks to the Author):

This is a revised submission of the previous version submitted by Misas-Villamil et al further examining the role of the Pit2 effector in *U. maydis*. I was generally quite satisfied with the original submission and I felt that it did significantly contribute to the field. I did though also feel that a few issues needed to be clarified (which I note some of which were also picked up by the other reviewers). I am happy to see that the authors have taken these concerns seriously and in my opinion, have adequately addressed each of these.

We want to thank our reviewers for improving our manuscript with their comments and suggestions.